# MuBench: Assessment of Multilingual Capabilities of Large Language Models Across 61 Languages

## Abstract

Multilingual large language models (LLMs) are advancing rapidly, with new models frequently claiming support for an increasing number of languages. However, existing evaluation datasets are limited and lack cross-lingual alignment, leaving assessments of multilingual capabilities fragmented in both language and skill coverage. To address this, we introduce MuBench, a benchmark covering 61 languages with 3.9M samples and evaluating a broad range of capabilities. We evaluate several state-of-the-art multilingual LLMs and find notable gaps between claimed and actual language coverage, particularly a persistent performance disparity between English and low-resource languages. Leveraging MuBench's alignment, we propose Multilingual Consistency (MLC) as a complementary metric to accuracy for analyzing performance bottlenecks and guiding model improvement. MuBench provides flexible evaluation formats, including mixed-language testing. Experimental results show that increasing model size does not improve its ability to handle mixed-language contexts. We recruited human experts to evaluate translation quality and cultural sensitivity for 34k samples across 17 languages, and combined these assessments with an LLM-as-a-Judge approach to ensure overall data quality in low resource languages. Our data is open at `https://huggingface.co/datasets/trustunogen/nYtVx4RmQp7wZc`

## 1 Introduction

Recent developments in large language models (LLMs) reflect a clear shift toward broad multilingual support. For instance, Gemma3 (Team, 2025) reports support for over 140 languages, while Qwen3 (Yang et al., 2025) emphasizes wide linguistic coverage across 119 languages and dialects. Proprietary models such as GPT-4o (OpenAI, 2024), Claude [1], and Gemini (Team, 2024) also highlight strong multilingual capabilities.

Despite rapid advances in multilingual LLMs, evaluating their capabilities across languages remains a core challenge. The multilingual evaluations in their technical reports cover only a small number of languages and a narrow range of capabilities (Yang et al., 2025). Moreover, multilingual evaluation involves more dimensions of assessment compared with single-language evaluation. Assessments should go beyond per-language task performance to include relative performance across languages, cross-lingual knowledge transfer (Lample & Conneau, 2019; Conneau et al., 2020), and robustness in mixed-language contexts (Chua et al., 2025; Huzaifah et al., 2024). Evaluation along these dimensions requires broad language and task coverage, as well as aligned test samples across languages. Existing multilingual benchmarks fall short in at least one of these aspects. Table 1 presents the comparison between popular multilingual benchmarks (INCLUDE (Romanou et al., 2024), MultiLoKo (Hupkes & Bogoychev, 2025), BenchMax (Huang et al., 2025) and MuBench).

To address these limitations, we introduce MuBench, a comprehensive multilingual benchmark spanning 61 languages and a diverse range of tasks, including natural language understanding, commonsense reasoning, factual recall, knowledge-based QA, academic and technical reasoning, and

---

[1] `https://www.anthropic.com/news/claude-4`

| Benchmark | Languages | Ability | Tasks | Samples | Cross-lingual Alignment | Multiple Formats | Code-switched Evaluation |
|-----------|-----------|---------|-------|---------|------------------------|------------------|--------------------------|
| INCLUDE | 44 | 1 | 1 | 22,655 | × | × | × |
| MultiLoKo | 31 | 1 | 1 | 15,500 | ✓ | × | × |
| BenchMax | 17 | 6 | 9 | 177,684 | ✓ | × | × |
| **MUBENCH** | **61** | **6** | **12** | **3,921,751** | ✓ | ✓ | ✓ |

Table 1: Comparison of multilingual benchmarks.

truthfulness. MUBENCH ensures cross-lingual alignment by maintaining consistent test items across languages, enabling fair and direct comparisons. We construct MUBENCH by translating widely used English benchmarks through an automated pipeline with rigorous quality control. We include code-switched variants that mix multiple languages within a single test item, allowing evaluation under multilingual input conditions. Cultural applicability is also assessed to remove items with obscure cultural references or Western-centric biases, mitigating cultural skew. Finally, stratified human evaluations across 17 languages validate the quality and fidelity of the translations.

Using MUBENCH, we conduct extensive evaluations of state-of-the-art LLMs and find that current models often fall short of their claimed multilingual coverage. A persistent performance gap remains between English and low-resource languages, and this gap does not consistently narrow with increased model size. In code-switched evaluation, we find that larger models do not necessarily exhibit greater robustness. Leveraging MUBENCH's fully aligned test samples, we analyze cross-lingual consistency and observe stable inter-language correlation patterns in each model, revealing implicit structures in multilingual knowledge sharing. We also investigate the impact of parallel corpora in pre-training on cross-lingual transfer of language abilities (Appendix C). These findings highlight the importance of analyzing the relationship between consistency and accuracy as a diagnostic tool for identifying multilingual performance bottlenecks—whether due to insufficient task knowledge or limited generalization across languages. MUBENCH thus provides a rigorous framework for understanding and advancing multilingual LLM development.

In summary, our contributions are:

**1)** We introduce MUBENCH, a multilingual benchmark supporting 61 languages that enables consistent and cross-lingual evaluation across 6 capabilities and 12 tasks.

**2)** We propose an automated data construction approach to reduce reliance on human annotation, enabling rapid scaling of multilingual evaluation. We design a rigorous quality-control pipeline that combines human evaluation with LLM-based evaluation.

**3)** We conduct extensive experiments to evaluate MUBENCH's utility, providing valuable insights into the strengths and weaknesses of existing multilingual LLMs, language influence pattern and mixed-language stability.

## 2 RELATED WORK

Several prior efforts have attempted to construct multilingual evaluation benchmarks. Local MMLU datasets like CMMLU (Li et al., 2024) and ArabicMMLU (Koto et al., 2024) collect data from local exams and across diverse educational levels and subjects. INCLUDE (Romanou et al., 2024) established an evaluation suite for local knowledge sourced from exams under a variety of regional contexts, supporting 44 languages. MultiLoKo (Hupkes & Bogoychev, 2025) extracts local documents in 31 languages from Wikipedia and organizes them into knowledge-based QA test questions. Those benchmarks only focus on knowledge QA capability and cannot constitute a comprehensive evaluation. They handle each language separately, without aligning the test samples across multiple languages. It causes fragmentation of the evaluation between languages. Moreover, these benchmarks rely entirely on manual annotation, making them difficult to scale further and leaving the gap between low-resource language evaluation and English evaluation unresolved.

In contrast to benchmarks built from native-language corpora, other efforts have extended high-quality English benchmarks into multiple languages. BenchMAX (Huang et al., 2025) extends 10 benchmark from English into 17 languages. BMLAMA (Qi et al., 2023) includes up to 53 languages with factual question answering task. GeoMLAMA (Yin et al., 2022) focuses on regional cultural differences, building in English and translating to another 4 languages. Other translation-based works include (Singh et al., 2025; Lin et al., 2022b; Lai et al., 2023; Xuan et al., 2025). These works either cover a limited range of evaluation capabilities or support too few languages to comprehensively assess the multilingual proficiency of today's LLMs. In addition, during the translation process, these works either rely heavily on human translation, which limits scalability, or use machine translation but the data construction procedure and quality control are not transparent.

## 3 MUBENCH

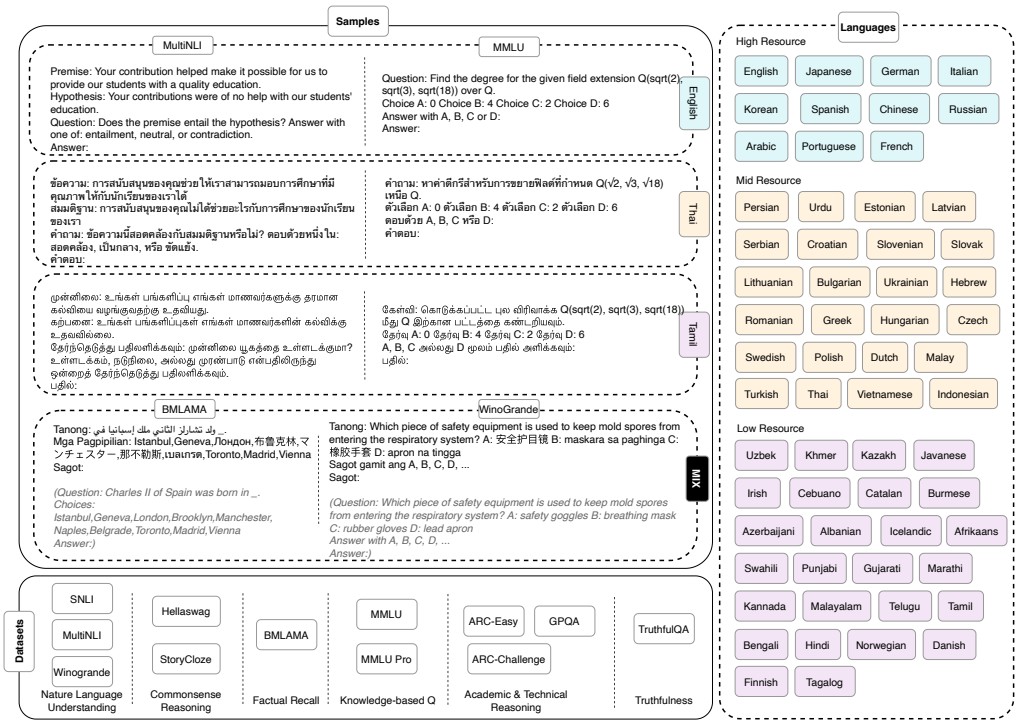

Figure 1: Overview of MUBENCH. MUBENCH supports 61 languages and covers popular datasets for evaluating natural language understanding, knowledge, and reasoning abilities. It also provides multiple variants for each dataset to accommodate different evaluation methods.

We extend widely-used English benchmarks to a broader set of languages while covering a diverse range of capabilities, including: Natural Language Understanding: SNLI (Bowman et al., 2015), MultiNLI (Williams et al., 2018) and WinoGrande (Sakaguchi et al., 2019); Commonsense Reasoning: HellaSwag (Zellers et al., 2019) and StoryCloze (Mostafazadeh et al., 2016); Factual Recall: BMLAMA (Qi et al., 2023); Knowledge-based QA: MMLU (Hendrycks et al., 2021) and MMLUPro (Wang et al., 2024); Academic & Technical Reasoning: GPQA (Rein et al., 2023), ARC-Easy and ARC-Challenge (Clark et al., 2018); Truthfulness: TruthfulQA (Lin et al., 2022a). This selection also spans a range of difficulty levels, from relatively simple datasets like StoryCloze to more challenging ones such as GPQA. For language selection, we chose the 61 most widely spoken languages based on the number of native speakers, covering over 60% of the global population (native speakers only) (Lis, 2025). Figure 1 illustrates the languages, data structure, and examples of MUBENCH.

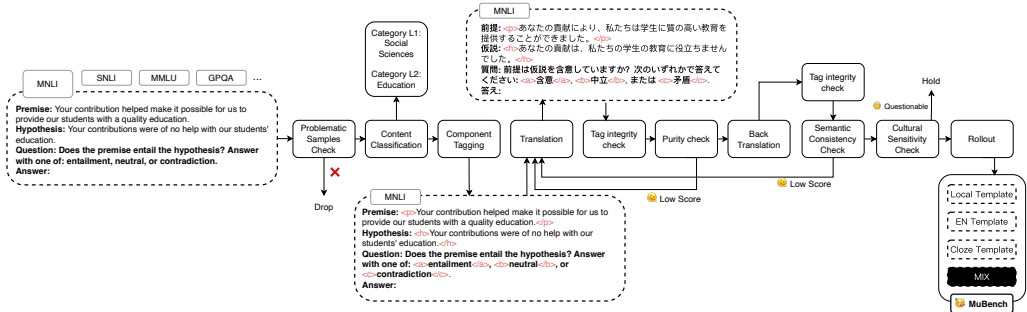

Figure 2: MUBENCH data collection pipeline. MUBENCH has established an automated benchmark translation framework with strict rules to control the quality. Each sample is labeled with content categories and undergoes a cultural sensitivity check.

## 3.1 DATA PIPELINE

We developed a rigorous data pipeline, as shown in Figure 2, comprising several main stages: **content classification**, **translation**, **semantic consistency evaluation**, **translation purity assessment**, and **cultural sensitivity check**. The finalized dataset variants constitute MUBENCH.

**Content Classification**   In addition to covering a broad spectrum of capabilities, MUBENCH also emphasizes sample-level diversity analysis. To achieve this, we extend the subject classification schema from MMLU by introducing additional categories that capture more everyday and real-world scenarios, structured in a two-level hierarchy. For each benchmark sample, GPT-4o is used to perform content-based classification—focusing on the topic rather than question type—by first selecting the most suitable high-level category, followed by a corresponding subcategory within it.

**Translation**   To preserve the structural consistency of test samples and enable future flexibility, we wrap each component of a question—such as the prompt and answer choices—with explicit tags and concatenate them into a unified text block for translation via GPT. Post-translation, we perform strict validation to ensure tag integrity; samples with missing or corrupted tags are flagged for retranslation. This design ensures the complete and faithful translation of the prompt, question stem, and answer choices. It also facilitates flexible modification of question formats in the future, allowing adaptation to different evaluation protocols tailored to various model types. Crucially, this design enables the construction of mixed-language test cases, allowing for targeted assessment of LLMs under code-switching and multilingual conditions.

**Semantic Consistency Evaluation**   At this stage, we control for semantic shifts introduced during translation. Each sample is first translated into the target language using GPT-4o, then back-translated into English. The original and back-translated English texts are compared, with GPT-4o assigning a semantic consistency score on a custom 1-to-5 scale. Samples receiving low scores (1 or 2) are flagged for retranslation. This procedure not only ensures semantic fidelity but also serves as a proxy for evaluating GPT-4o's translation performance in low-resource languages.

**Translation Purity Assessment**   Maintaining semantic consistency alone is insufficient; translations must also exhibit linguistic authenticity in the target language and avoid inappropriate English intrusions. While the retention of certain English proper nouns may be acceptable, we prioritize replacing them with widely recognized equivalents in the target language to ensure natural and native-like expression. To evaluate this, we define a 1-to-5 scoring rubric and prompt GPT to assess the linguistic purity of each translation.

**Cultural Sensitivity Checking**   Finally, It is essential to ensure that a question, once translated into the target language, remains culturally appropriate and does not conflict with the cultural context of that language. Commonsense knowledge can vary significantly across cultures, potentially altering the correct answer if cultural assumptions shift during translation. To address this, we design a

prompt that instructs GPT-4o to identify and annotate instances of cultural shift in the translated samples.

**Rollout** We construct several variants for each tasks. **Local Template**: Uses the native-language prompt and content to assess the model's ability to follow instructions and answer within the linguistic context of the target language. **EN Template**: Keeps the sample content in the target language but uses the English prompt. This format aligns with many existing multilingual benchmarks and often leads to improved performance due to models' stronger instruction-following capabilities in English. **Cloze Template** (Alzahrani et al., 2024; Clark et al., 2018): Removes explicit task

Table 2: Sample statistics of MUBENCH. **CS** stands for Culturally Sensitive

| Dataset | Origin Samples | CS Samples | Final Samples |
|---|---|---|---|
| SNLI | 613,050 | 5,314 | 549,000 |
| MultiNLI | 602,802 | 4,091 | 541,924 |
| StoryCloze | 95,221 | 2,522 | 81,252 |
| WinoGrande | 80,322 | 220 | 76,860 |
| BMLAMA | 413,831 | 1,125 | 369,721 |
| MMLU | 873,946 | 18,058 | 768,112 |
| MMLU Pro | 738,212 | 5,302 | 696,315 |
| HellaSwag | 615,534 | 8,331 | 554,368 |
| ARC-Easy | 147,986 | 72 | 146,949 |
| ARC-Challenge | 74,542 | 28 | 74,054 |
| GPQA | 27,328 | 0 | 27,328 |
| TruthfulQA | 49,837 | 3,149 | 35,868 |
| **Total** | **4,332,611** | **48,212** | **3,921,751** |

instructions and instead organizes the question and answer choices into natural sentences. Model performance is evaluated based on which option yields the lowest perplexity (PPL). This format is particularly effective for early-stage or smaller models that may struggle with instruction comprehension. **MIX**: For each of the above variants, we additionally construct a code-switched version by randomly replacing components (e.g., prompt, options) with content in another language at a controlled probability, allowing robust testing under mixed-language settings.

## 3.2 DATA ANALYSIS

**Statistics** Table 2 presents the number of samples included in each dataset within MUBENCH, which constitutes a significantly larger scale than previous dataset expansion efforts. During the final rollout, we removed samples flagged for cultural sensitivity, as well as those receiving the lowest scores in semantic consistency and linguistic purity evaluations. Moreover, all languages are aligned; thus, if a sample is filtered out in one language, its counterparts in all other languages are also removed accordingly. More details of cultural sensitive samples and the diversity are present in the appendix.

## 3.3 QUALITY CONTROL AND HUMAN EVALUATION

During dataset translation, samples scoring below 3 in either semantic consistency or linguistic purity were retranslated multiple times.

We conducted human evaluations on 2,000 samples per language across 17 languages, using the same scoring criteria. Additionally, 100 matched samples from 9 languages in OpenAI MMMLU[2] and MUBENCH were evaluated to directly compare GPT-4o translations with human ones.

Table 3 shows that human scores for MUBENCH and OpenAI MMMLU are closely aligned, with no significant difference across 8 of 9 languages; the only exception is Chinese, where MUBENCH shows slightly lower consistency. Table 4 compares GPT-4o's self-assessments with human scores, revealing that GPT-4o tends to underrate its translations, indicating conservative scoring. Overall, MUBENCH achieves translation quality on par with human-translated benchmarks. The detail of consistency and purity distribution are included in the appendix.

For translation quality details, we report COMET scores and GPT-4o consistency scores in Table 5 by three language tiers on terminology-dense Mubench datasets. On low resource languages, GPT's consistency scores also remain at a very high level. Through our analysis of COMET scores, we observed that the COMET score significantly dropped for certain languages, such as Cebuano (ceb) with a score of $0.5858 \pm 0.0543$. However, manual spot checks did not reveal a corresponding drop in translation quality. This suggests that the COMET model's limited support for low-resource languages may also contribute to the lower scores. In summary, MuBench maintains high data quality even for low-resource languages. More details of human evaluation is elaborated in Appendix A.6.

---

[2]https://huggingface.co/datasets/openai/MMMLU

Table 3: Per-language comparison of Semantic Consistency and Translation Purity between OpenAI-MMMLU and MᴜBᴇɴᴄʜ-MMLU (mean scores only, with $t$-test $p$-values).

| Lang | $n$ | Semantic Consistency | | | Translation Purity | | |
|------|-----|------|------|------|------|------|------|
| | | MMMLU | Ours | $p$ | MMMLU | Ours | $p$ |
| es | 100 | 4.91 | 5.00 | 0.0061 | 4.93 | 4.98 | 0.1667 |
| ja | 100 | 4.13 | 4.24 | 0.1803 | 3.73 | 3.81 | 0.2188 |
| pt | 100 | 4.84 | 4.89 | 0.3718 | 4.94 | 4.94 | 1.0000 |
| ko | 100 | 4.73 | 4.78 | 0.5663 | 4.51 | 4.45 | 0.5471 |
| it | 100 | 4.79 | 4.76 | 0.7075 | 4.94 | 4.97 | 0.3197 |
| id | 100 | 4.95 | 4.93 | 0.5298 | 4.83 | 4.88 | 0.2534 |
| de | 100 | 5.00 | 4.95 | 0.1324 | 5.00 | 5.00 | – |
| zh | 100 | 4.31 | 3.85 | 0.0000 | 4.69 | 4.79 | 0.0584 |
| fr | 100 | 5.00 | 5.00 | – | 4.98 | 4.96 | 0.4823 |
| ar | 100 | 5.00 | 5.00 | – | 4.85 | 4.82 | 0.5343 |
| All | 900 | 4.74 | 4.71 | 0.2980 | 4.73 | 4.75 | 0.3700 |

Table 4: Per-language GPT vs Human ratings on Semantic Consistency and Translation Purity (mean ± std).

| Lang | Semantic Consistency | | Translation Purity | |
|------|------|------|------|------|
| | Human | GPT | Human | GPT |
| th | 4.865±0.431 | 3.887±1.267 | 4.805±0.577 | 3.717±1.200 |
| es | 4.947±0.279 | 4.107±1.154 | 4.926±0.314 | 3.826±1.227 |
| fr | 4.994±0.092 | 4.189±1.135 | 4.903±0.344 | 3.777±1.143 |
| vi | 4.836±0.504 | 3.956±1.269 | 4.603±0.812 | 3.844±1.229 |
| tr | 4.781±0.596 | 3.953±1.269 | 4.614±0.761 | 3.824±1.174 |
| id | 4.859±0.411 | 4.173±1.146 | 4.748±0.461 | 3.668±1.235 |
| tl | 4.738±0.572 | 4.035±1.218 | 4.681±0.581 | 3.364±1.276 |
| ko | 4.674±0.740 | 3.949±1.277 | 4.569±0.888 | 3.883±1.176 |
| pt | 4.774±0.598 | 4.125±1.146 | 4.776±0.624 | 3.974±1.212 |
| nl | 4.805±0.554 | 4.176±1.171 | 4.777±0.517 | 3.739±1.247 |
| it | 4.774±0.580 | 4.189±1.131 | 4.782±0.558 | 3.798±1.235 |
| ru | 4.729±0.600 | 4.179±1.153 | 4.761±0.534 | 3.975±1.167 |
| de | 4.860±0.409 | 4.355±1.055 | 4.828±0.432 | 3.755±1.209 |
| zh | 4.358±0.738 | 4.045±1.190 | 4.739±0.492 | 3.931±1.166 |
| ja | 4.104±0.655 | 4.150±1.122 | 3.623±0.765 | 4.015±1.061 |
| ar | 4.995±0.071 | 4.014±1.249 | 4.784±0.412 | 3.626±1.217 |

Table 5: Translation quality (COMET / GPT-4o semantic consistency scores) by language tier.

| Tier | ARCChallenge | ARCEasy | MMLU | GPQA | TruthfulQA |
|------|------|------|------|------|------|
| High | 85.9±1.5 / 4.84±0.04 | 85.8±1.7 / 4.81±0.03 | 82.4±2.0 / 4.79±0.05 | 79.8±2.0 / 4.87±0.08 | 85.7±1.3 / 4.82±0.05 |
| Mid | 86.2±1.3 / 4.81±0.04 | 86.1±1.6 / 4.78±0.04 | 83.0±1.5 / 4.76±0.05 | 79.8±1.8 / 4.88±0.04 | 86.2±1.2 / 4.80±0.05 |
| Low | 83.4±5.4 / 4.64±0.22 | 83.2±5.2 / 4.58±0.25 | 80.2±5.5 / 4.61±0.23 | 77.4±4.4 / 4.75±0.18 | 84.2±4.3 / 4.69±0.15 |

# 4 MULTILINGUAL CAPABILITY EVALUATION

## 4.1 OVERVIEW

Since the pretraining stage plays a crucial role in determining the multilingual capabilities of large language models (LLMs), our evaluation focuses on the base versions of various model families. While MᴜBᴇɴᴄʜ is designed with the flexibility to adapt test samples to different task formats, we mainly focus on its application to the base models. Importantly, MᴜBᴇɴᴄʜ allows for evaluations of chat-oriented models by providing instructions tailored to each language.

We perform zero-shot evaluations on **Qwen3** (Yang et al., 2025), **Qwen2.5** (Qwen et al., 2025), **Gemma2** (Team et al., 2024), and **Gemma3** (Team, 2025) models ranging from 1–3B, 7–14B, up to 70B. Babel (Zhao et al., 2025) series are also included, which are built upon Qwen2.5 models and aims to cover the top 25 most widely spoken languages. Moreover, dedicated for 13 SouthEast Asian (SEA) languages, Sailor2 (Dou et al., 2025) series is also Qwen2.5-like models and we include them into the comparison. The evaluation is conducted using MᴜBᴇɴᴄʜ **cloze template** variants. An exception is made for **SNLI** and **MultiNLI**, where we adopt the **local template** in a QA-style with 10-shot settings. We report accuracy (**ACC**) on SNLI, MultiNLI, WinoGrande, and BMLAMA, and char-length normalized accuracy (**ACC_NORM**) on the other datasets. Additionally, we also evaluated **GPT-4o**. Since it is not a base model, we assessed its performance on each benchmark using **local template** and report Exact Match (EM) scores.

Table 6 summarizes the performance of selected LLMs on MᴜBᴇɴᴄʜ, along with their performance gaps relative to English. While GPT-4o substantially outperforms open-source base models across the board (noting that evaluation protocols differ), it still exhibits a clear drop in performance for non-English languages.

Among open models, Qwen demonstrates strong and consistent performance across a wide range of tasks. This is particularly evident in inference-focused benchmarks (MultiNLI), knowledge-intensive tasks (BMLAMA, MMLU), and QA-style datasets (ARC). Both Qwen3-14B and Qwen2.5-72B stand out for their balanced and robust performance across nearly all evaluation metrics. In contrast, Gemma models—especially Gemma-3-27B-pt—excel in narrative and com-

Table 6: Performance of LLMs on MUBENCH. The values in parentheses indicate the score differences relative to English performance.

| | MNLI | StoryCloze | WinoGrande | BMLAMA | MMLU | HellaSwag | ARCEasy | ARCChallenge |
|---|---|---|---|---|---|---|---|---|
| *Proprietary Model* | | | | | | | | |
| gpt-4o-2024-05-13 | 69.78 (-11.18) | 97.68 (-1.62) | 71.68 (-10.35) | 66.87 (-6.90) | 70.01 (-2.26) | 83.02 (-10.75) | 93.64 (-5.00) | 87.32 (-7.35) |
| *Model (1–4B)* | | | | | | | | |
| Qwen3-0.6B-Base | 38.45 (-30.53) | 56.05 (-15.78) | 50.67 (-6.20) | 27.17 (-32.19) | 26.88 (-5.38) | 31.01 (-21.29) | 29.75 (-19.25) | 24.62 (-8.89) |
| Qwen3-1.7B-Base | 56.33 (-24.75) | 59.71 (-17.84) | 50.99 (-6.30) | 31.89 (-28.45) | 28.13 (-7.30) | 35.68 (-28.29) | 33.46 (-23.00) | 26.88 (-9.80) |
| Qwen3-4B-Base | 69.26 (-4.47) | 64.16 (-17.19) | 53.27 (-10.04) | 37.82 (-26.87) | 30.18 (-8.38) | 42.52 (-29.57) | 37.55 (-19.51) | 30.09 (-9.43) |
| Qwen2.5-0.5B | 35.10 (-25.94) | 54.26 (-17.10) | 50.39 (-3.44) | 26.42 (-39.55) | 26.27 (-4.85) | 29.42 (-20.54) | 28.06 (-21.83) | 23.67 (-7.34) |
| Sailor2-1B | 34.56 (+2.06) | 54.82 (-18.32) | 49.98 (-5.50) | 28.37 (-37.95) | 26.22 (-3.45) | 29.88 (-20.30) | 28.83 (-18.18) | 23.51 (-5.79) |
| Qwen2.5-1.5B | 46.11 (-29.98) | 56.17 (-24.63) | 50.48 (-10.94) | 31.91 (-37.04) | 27.19 (-7.73) | 31.64 (-33.95) | 29.51 (-24.67) | 24.62 (-12.92) |
| gemma-3-1b-pt | 32.66 (+0.22) | 56.91 (-10.74) | 51.62 (-5.76) | 41.71 (-27.31) | 26.62 (-1.29) | 31.11 (-13.02) | 28.94 (-7.77) | 24.84 (-2.05) |
| gemma-3-4b-pt | 42.48 (-5.82) | 58.31 (-9.65) | 56.01 (-11.43) | 52.57 (-17.96) | 26.70 (-1.40) | 34.31 (-16.81) | 29.26 (-10.08) | 24.47 (-2.94) |
| gemma-2-2b | 34.51 (-12.74) | 63.98 (-18.91) | 52.53 (-11.94) | 40.48 (-30.73) | 28.05 (-6.27) | 40.29 (-30.46) | 33.45 (-16.53) | 27.36 (-8.81) |
| *Model (7–20B)* | | | | | | | | |
| Qwen3-8B-Base | 76.16 (-6.56) | 67.87 (-16.42) | 55.41 (-12.03) | 47.44 (-24.70) | 31.47 (-8.14) | 47.72 (-28.02) | 40.51 (-17.90) | 31.73 (-8.13) |
| Qwen3-14B-Base | 81.63 (-0.92) | 71.14 (-13.61) | 57.67 (-15.04) | 51.72 (-21.14) | 32.61 (-8.22) | 52.86 (-25.90) | 42.75 (-15.41) | 33.71 (-5.98) |
| Qwen2.5-7B | 67.23 (-18.14) | 61.88 (-22.02) | 51.68 (-14.68) | 36.02 (-28.39) | 29.77 (-9.56) | 39.52 (-36.92) | 35.49 (-24.49) | 28.14 (-11.98) |
| Sailor2-8B | 54.66 (-25.99) | 61.89 (-20.62) | 52.59 (-11.96) | 40.26 (-30.47) | 28.25 (-7.76) | 38.44 (-34.76) | 34.11 (-22.44) | 26.62 (-11.01) |
| Babel-9B | 66.38 (-22.27) | 61.96 (-21.48) | 53.29 (-14.72) | 42.73 (-29.34) | 29.15 (-9.30) | 40.57 (-34.25) | 34.25 (-27.73) | 27.64 (-13.08) |
| Qwen2.5-14B | 74.24 (-11.83) | 66.50 (-19.26) | 50.19 (-11.89) | 23.68 (-31.04) | 31.64 (-9.70) | 45.62 (-35.09) | 39.05 (-20.59) | 31.20 (-11.07) |
| Sailor2-20B | 73.36 (-16.07) | 67.41 (-18.50) | 56.30 (-18.64) | 48.11 (-25.13) | 30.61 (-8.94) | 46.74 (-32.83) | 38.14 (-20.95) | 30.36 (-10.71) |
| gemma-3-12b-pt | 37.08 (-4.45) | 55.42 (-4.02) | 61.40 (-11.56) | 59.61 (-12.17) | 26.27 (-0.87) | 30.50 (-4.01) | 28.27 (-3.40) | 24.23 (+1.21) |
| gemma-2-9b | 65.10 (-12.05) | 73.40 (-12.28) | 57.98 (-13.83) | 53.59 (-18.12) | 31.64 (-7.18) | 55.66 (-22.19) | 41.75 (-13.91) | 33.27 (-7.11) |
| *Model (>20B)* | | | | | | | | |
| Qwen2.5-32B | 80.36 (-7.61) | 68.19 (-18.57) | 56.95 (-17.91) | 48.84 (-23.45) | 33.30 (-8.51) | 49.43 (-32.07) | 41.51 (-17.96) | 33.12 (-10.95) |
| Qwen2.5-72B | 84.48 (-5.53) | 71.89 (-15.42) | 59.17 (-18.82) | 52.87 (-19.79) | 36.25 (-7.59) | 54.99 (-28.77) | 46.73 (-15.50) | 36.40 (-9.13) |
| Babel-83B | 85.29 (-5.04) | 71.40 (-15.83) | 58.52 (-18.89) | 52.46 (-20.91) | 34.75 (-8.19) | 54.65 (-28.33) | 43.08 (-18.51) | 34.47 (-8.06) |
| gemma-3-27b-pt | 77.12 (-8.60) | 79.06 (-8.48) | 63.49 (-13.34) | 61.74 (-10.48) | 36.46 (-4.84) | 66.09 (-14.28) | 48.18 (-7.01) | 37.99 (-3.59) |
| gemma-2-27b | 75.38 (-8.58) | 77.21 (-10.17) | 60.78 (-15.81) | 56.09 (-14.85) | 34.09 (-6.76) | 62.08 (-20.02) | 44.23 (-9.48) | 35.70 (-3.90) |

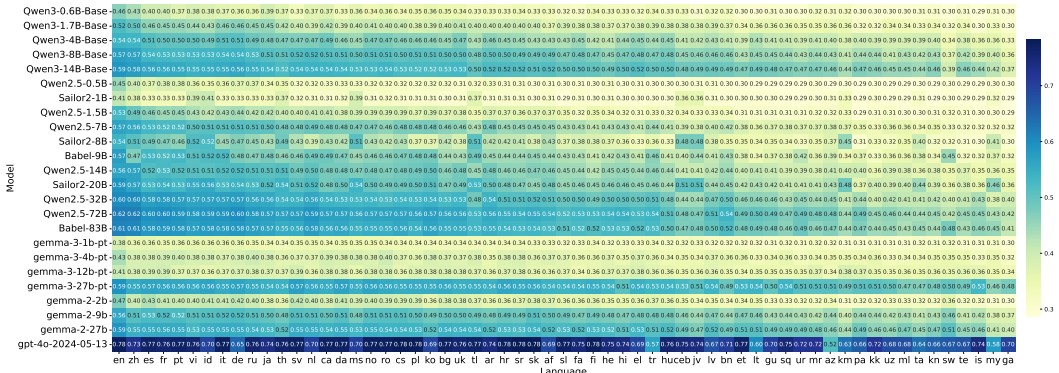

Figure 3: Model performance by language.

monsense reasoning tasks such as StoryCloze and HellaSwag, and perform competitively on factual knowledge benchmarks like BMLAMA. Overall, Qwen models offer stronger and more stable performance, whereas Gemma exhibits sharper peaks in specific reasoning-heavy tasks. Both Babel-9B and Sailor2-8B are extended from Qwen2.5-7B. Babel-9B generally retains the capabilities of its base model, with modest gains in factual QA and language understanding tasks (e.g., BMLAMA, WinoGrande). In contrast, Sailor2-8B shows a broad regression, suggesting that its specialized training on Southeast Asian languages may have compromised its performance on other languages. Notably, Babel-83B underperforms relative to its baseline Qwen2.5-72B, despite a larger parameter count, with performance degradation particularly evident on knowledge-heavy tasks such as MMLU and ARC.

As expected, larger models tend to achieve better overall performance. However, the relative performance gap between English and other languages does not consistently narrow with scale. This trend holds across most tasks, with the exception of SNLI. These findings suggest that the performance gap for low-resource languages remains persistent, and only begins to close when a model approaches saturation in English performance on a given benchmark. The evaluation results on full MUBENCH test sets are presented in Appendix E.

## 4.2 PER-LANGUAGE PERFORMANCE COMPARISON

Figure 3 presents the per-language performance of the evaluated LLMs, measured as the mean score across all datasets for each language. As expected, models tend to perform better on high-resource languages such as English, Chinese, and Spanish, while lower-resource languages generally yield lower scores. GPT-4o demonstrates strong multilingual performance across all 61 languages. However, when normalized against its own English performance, notable drops are observed in languages such as Tagalog (tl) and Burmese (my). Interestingly, several affected languages—such as Chinese (zh) and German (de)—are not traditionally considered low-resource, underscoring the broader challenges in achieving consistent performance across typologically and culturally diverse languages.

Among open-source models, Gemma-3-27B emerges as the best overall performer, achieving consistently strong results across nearly all languages. The Babel and Sailor2 models demonstrate notable gains in their targeted language groups, though often at the expense of reduced performance in others. Larger models from the Qwen2.5 and Qwen3 series also perform well, with performance improving steadily with increased model size. These findings highlight the critical role of both scale and model design in achieving robust and balanced multilingual capabilities.

## 4.3 CROSS-LINGUAL CONSISTENCY EVALUATION

Evaluating multilingual LLMs goes beyond per-language accuracy. Consistency across languages—producing similar responses even when incorrect—signals shared cross-lingual representations and potential for improvement. As such, consistency serves as a crucial complement to accuracy. Qi et al. (Qi et al., 2023) introduced BMLAMA to evaluate cross-lingual consistency using ranking-based scores. However, for multiple-choice questions with discrete answers (e.g., "What is the capital of China?"), only the top choice matters—ranking secondary options is often irrelevant and may distort consistency assessment. We instead use a multilingual consistency (MLC) metric based on exact Top-1 answer match across languages: $\text{MLC}(l, l') = \frac{1}{|N|} \sum_{i=1}^{N} \mathbf{1}_{c_i = c_i'}$, where $N$ is the number of questions, $l$ and $l'$ are two languages, and $c_i$, $c_i'$ are the model's Top-1 choices for the same question in $l$ and $l'$, respectively. All MUBENCH samples are aligned across 61 languages, providing a robust foundation for consistent cross-lingual evaluation and analysis of knowledge transfer.

Table 7 reports average MLC scores across all language pairs, and between each language and English. In general, MLC correlates with accuracy—models with higher accuracy tend to exhibit better consistency. However, notable exceptions reveal important dynamics. For example, in MultiNLI, GPT-4o achieves lower accuracy than several open-source models above 20B parameters, yet maintains competitive or superior consistency (e.g., outperforming gemma-2-27b), suggesting stronger cross-lingual representation alignment. Conversely, in MMLUPro and GPQA, GPT-4o significantly outperforms gemma-3-27b-pt and gemma-2-27b in accuracy, but lags in consistency, indicating less overlap in correct answers across languages. These discrepancies highlight that accuracy and consistency reflect distinct facets of multilingual performance. Low consistency suggests fragmented cross-lingual representations, while low accuracy indicates limited task knowledge. We therefore advocate using MLC alongside accuracy to better diagnose model weaknesses and inform multilingual model development. Additionally, we find that consistency between each language and English is generally higher than the average across all language pairs, reaffirming English's central role in multilingual LLMs.

## 4.4 CROSS-LINGUAL INFLUENCE PATTERN

Beyond measuring overall consistency, MLC scores also reveal patterns of cross-lingual interaction within LLMs. Figure 4 visualizes these interactions on the BMLAMA task, with 61 languages grouped by family and ordered by resource availability. Each cell represents the consistency score between a language pair. Since consistency is influenced by accuracy, to isolate language interaction patterns independent of accuracy, we normalize MLC scores by the average accuracy of each pair: $\text{Rel-MLC}(l, l') = \frac{\text{MLC}(l, l')}{\text{Mean}(\text{ACC}_l, \text{ACC}_{l'})}$. We observe similar patterns across different models. Strong intra-family consistency is evident, especially within Germanic, Romance, Slavic, Indo-Aryan, Austronesian, and Dravidian families. Some pairs, like Croatian (hr) and Serbian (sr), show exceptionally high alignment. Notably, cross-family consistency—especially involving English and other Indo-European languages—extends to most language families, including isolates.

Table 7: Consistency across languages. 'All' refers to the average consistency across all language pairs, while 'vs. EN' indicates the average consistency between each language and English.

| | MNLI | | BMLAMA | | MMLU | | MMLUPro | | GPQA | | ARCEasy | | ARCChallenge | |
|---|---|---|---|---|---|---|---|---|---|---|---|---|---|---|
| | all | vs. EN | all | vs. EN | all | vs. EN | all | vs. EN | all | vs. EN | all | vs. EN | all | vs. EN |
| *Proprietary Model* | | | | | | | | | | | | | | |
| gpt-4o-2024-05-13 | 74.60 | 79.25 | 66.21 | 74.67 | 68.42 | 69.71 | 42.46 | 47.07 | 47.46 | 43.93 | 90.34 | 94.28 | 84.52 | 89.24 |
| *Model (1–4B)* | | | | | | | | | | | | | | |
| Qwen3-0.6B-Base | 49.51 | 51.04 | 29.64 | 35.36 | 49.22 | 48.98 | 44.84 | 42.07 | 64.00 | 63.52 | 39.42 | 40.44 | 40.94 | 41.26 |
| Qwen3-1.7B-Base | 56.72 | 62.92 | 33.92 | 42.06 | 49.82 | 50.21 | 44.14 | 42.48 | 64.00 | 64.70 | 41.45 | 43.91 | 42.66 | 43.48 |
| Qwen-4B-Base | 70.39 | 70.99 | 36.24 | 44.74 | 51.08 | 52.36 | 44.42 | 43.41 | 64.16 | 65.36 | 43.54 | 46.92 | 44.13 | 45.65 |
| Qwen2.5-0.5B | 42.98 | 48.39 | 27.93 | 34.21 | 47.67 | 45.94 | 45.06 | 40.15 | 62.83 | 60.68 | 37.17 | 37.19 | 39.40 | 38.03 |
| Sailor2-1B | 58.48 | 71.72 | 28.96 | 36.57 | 48.64 | 48.28 | 46.54 | 43.36 | 63.88 | 62.98 | 38.56 | 39.51 | 40.45 | 40.46 |
| Qwen2.5-1.5B | 45.29 | 55.07 | 32.64 | 40.60 | 48.31 | 47.52 | 43.53 | 39.96 | 63.44 | 61.53 | 38.52 | 39.92 | 39.87 | 38.64 |
| gemma-3-1b-pt | 86.21 | 92.58 | 40.64 | 51.17 | 52.79 | 54.52 | 48.13 | 48.67 | 65.85 | 67.30 | 40.81 | 43.46 | 42.78 | 43.76 |
| gemma-3-4b-pt | 42.04 | 44.77 | 51.35 | 61.04 | 50.89 | 53.02 | 45.52 | 45.80 | 64.08 | 64.18 | 39.47 | 42.46 | 41.23 | 42.81 |
| gemma-2-2b | 41.50 | 29.26 | 39.24 | 49.81 | 53.82 | 54.36 | 48.60 | 47.53 | 67.84 | 68.35 | 43.21 | 46.23 | 44.09 | 45.98 |
| *Model (7–20B)* | | | | | | | | | | | | | | |
| Qwen3-8B-Base | 74.47 | 78.24 | 45.48 | 55.63 | 51.39 | 52.52 | 44.59 | 43.79 | 64.79 | 66.12 | 45.23 | 48.91 | 45.21 | 46.85 |
| Qwen3-14B-Base | 80.76 | 79.75 | 49.80 | 59.66 | 52.59 | 54.06 | 45.02 | 44.74 | 64.85 | 66.84 | 46.26 | 49.78 | 46.01 | 48.45 |
| Qwen2.5-7B | 65.37 | 74.53 | 34.49 | 42.58 | 49.28 | 50.29 | 42.92 | 42.04 | 61.89 | 62.62 | 41.29 | 45.31 | 41.33 | 43.29 |
| Sailor2-8B | 49.86 | 60.37 | 38.36 | 48.17 | 50.40 | 50.37 | 44.98 | 43.13 | 64.89 | 64.63 | 42.04 | 44.21 | 42.07 | 43.57 |
| Babel-9B | 58.75 | 69.49 | 40.97 | 51.46 | 46.99 | 49.04 | 40.81 | 40.83 | 61.82 | 63.79 | 39.53 | 44.21 | 39.66 | 42.06 |
| Qwen2.5-14B | 74.97 | 79.11 | 26.19 | 31.21 | 50.08 | 51.71 | 43.35 | 42.79 | 62.96 | 64.29 | 43.20 | 47.79 | 42.89 | 45.41 |
| Sailor2-20B | 73.25 | 78.18 | 46.03 | 56.05 | 50.96 | 51.85 | 45.07 | 44.40 | 65.84 | 68.21 | 44.16 | 47.80 | 44.43 | 46.68 |
| gemma-3-12b-pt | 47.53 | 59.61 | 58.73 | 66.71 | 48.36 | 50.23 | 42.68 | 43.66 | 60.44 | 60.74 | 36.69 | 39.52 | 38.79 | 39.74 |
| gemma-2-9b | 70.00 | 74.91 | 51.62 | 61.12 | 55.87 | 57.51 | 50.12 | 49.77 | 71.00 | 73.30 | 47.12 | 51.71 | 47.02 | 49.58 |
| *Model (¿20B)* | | | | | | | | | | | | | | |
| Qwen2.5-32B | 80.83 | 84.48 | 46.54 | 56.12 | 50.91 | 52.75 | 43.07 | 43.12 | 61.88 | 63.80 | 44.21 | 48.69 | 43.74 | 47.11 |
| Qwen2.5-72B | 84.65 | 88.06 | 50.23 | 59.90 | 53.01 | 55.39 | 45.08 | 45.26 | 64.67 | 66.63 | 47.44 | 52.25 | 45.83 | 49.05 |
| Babel-83B | 85.20 | 88.34 | 50.17 | 59.73 | 52.70 | 55.09 | 45.46 | 45.64 | 65.66 | 66.39 | 46.24 | 50.90 | 45.59 | 48.59 |
| gemma-3-27b-pt | 77.43 | 82.09 | 61.02 | 68.10 | 58.66 | 61.91 | 53.24 | 54.88 | 73.72 | 74.46 | 52.16 | 55.87 | 51.07 | 54.29 |
| gemma-2-27b | 74.24 | 77.78 | 53.65 | 62.81 | 55.39 | 58.03 | 47.98 | 48.62 | 66.33 | 68.82 | 48.27 | 51.44 | 48.06 | 51.77 |

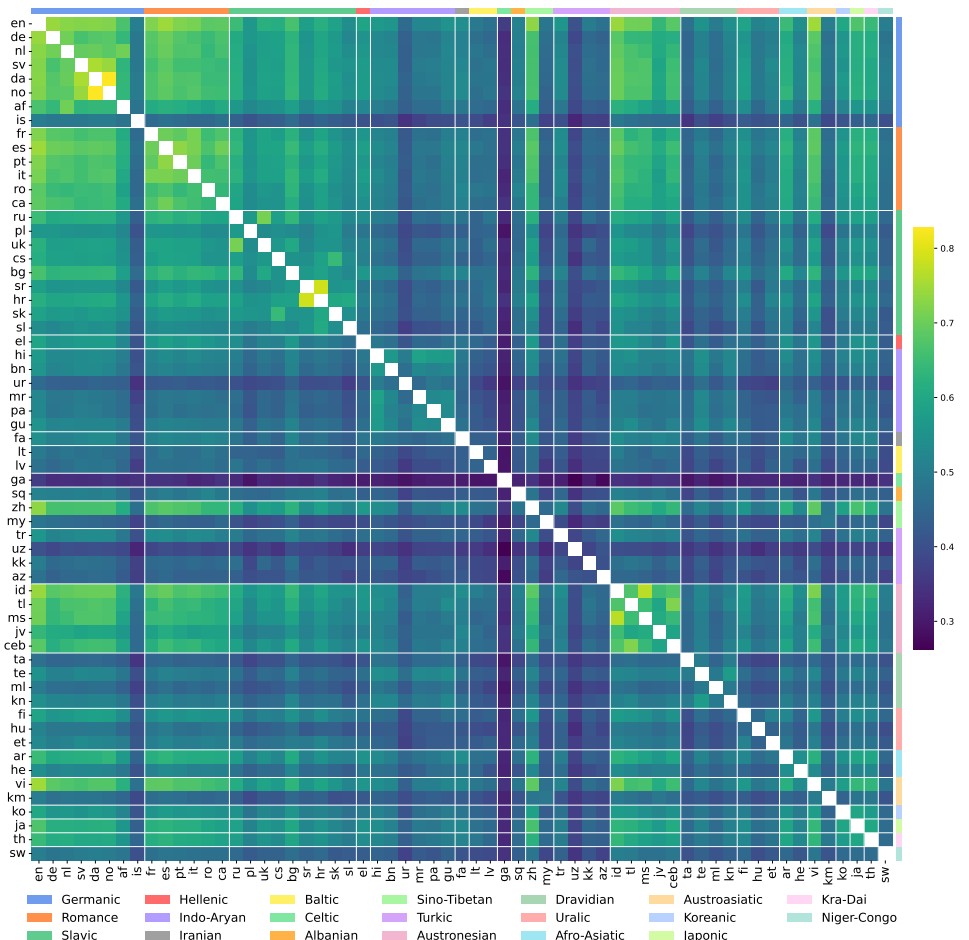

Figure 4: Consistency of Qwen3-14B-Base across languages tested on BMLAMA.

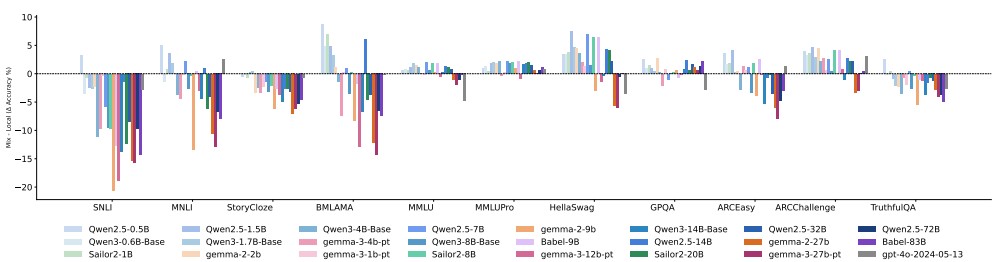

Figure 5: Model performance under mixed-language context.

These stable patterns reflect the underlying distribution of multilingual training data, rather than specific model architectures. Appendix B presents consistency results according to other linguistic typologies. We find that cross-lingual influence generally occurs within language families and is largely independent of morphological type or word order. Understanding such pattern of cross-lingual influence can provide guidance for configuring training data in multilingual LLMs.

### 4.5 PERFORMANCE UNDER CODE-SWITCHED CONTEXTS

An often overlooked aspect of multilingual LLMs is their ability to process and remain stable in mixed-language contexts. Chua et al. (Chua et al., 2025) identified a cross-lingual knowledge barrier in large models. Leveraging MUBENCH, we examine LLM behavior under such scenarios across a wide range of tasks by randomly replacing the template, question stem, and answer choices of each English test sample with other languages at a 0.5 probability. BMLAMA samples may contain up to 9 languages, while other benchmarks include up to 3 per sample.

Figure 5 shows the performance gap between the mixed-language setting and the average score across individual languages. The Qwen series exhibits greater stability in code-switched contexts compared to the gemma models. Interestingly, smaller models often benefit from the presence of English in mixed-language inputs, resulting in higher scores relative to their monolingual average. However, as model size increases, the gap between mixed-language performance and single-language gains widens—suggesting that improvements in multilingual understanding do not necessarily translate to better handling of mixed inputs.

These findings highlight the need to treat mixed-language performance as a distinct evaluation target. While LLMs may improve across individual languages, their ability to generalize under code-switching remains limited.

## 5 CONCLUSION

We present MUBENCH, a comprehensive multilingual benchmark for evaluating large language models (LLMs) across 61 languages. Through rigorous translation quality control and cross-lingual consistency evaluation, MUBENCH provides valuable insights into the strengths and limitations of current multilingual models. Our experiments highlight performance gaps between high-resource and low-resource languages, emphasizing the challenges in achieving consistent cross-lingual capabilities. This work offers a standardized tool for assessing multilingual LLMs and guides future improvements, particularly for low-resource languages. MUBENCH focuses only on evaluating knowledge that is universal across languages. However, another important aspect of multilingual evaluation is assessing language-specific, localized abilities, which will be a direction for our future work.

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

## A    DETAILS OF MUBENCH

### A.1    LANGUAGE SUPPORT

Table 8 presents the languages supported by MUBENCH. We rank the languages by their estimated number of native speakers, using data from Wikipedia[3] and other reputable online sources. To estimate the distribution of each language in web-scale data, we also report the number of tokens per language in the Common Crawl corpus. For this, we randomly selected one snapshot from each year between 2022 and 2024 and computed the average token proportion for each language. Considering only native speakers, these languages cover over 60% of the global population. When including second-language speakers, the coverage exceeds 99% worldwide.

Table 8: Languages sorted by native speakers and ratios in Common Crawl (HIGH at left, MID center, LOW right)

| Code | Name | Speakers | Tokens | Code | Name | Speakers | Tokens | Code | Name | Speakers | Tokens |
|---|---|---|---|---|---|---|---|---|---|---|---|
| zh | Chinese | 1390M | 6.34% | vi | Vietnamese | 86M | 1.35% | hi | Hindi | 345M | 0.31% |
| es | Spanish | 484M | 4.14% | tr | Turkish | 85M | 0.98% | bn | Bengali | 242M | 0.18% |
| ar | Arabic | 411M | 0.78% | ms | Malay | 82M | 0.03% | mr | Marathi | 83M | 0.04% |
| en | English | 390M | 42.62% | ur | Urdu | 78M | 0.04% | te | Telugu | 83M | 0.03% |
| pt | Portuguese | 250M | 1.51% | id | Indonesian | 75M | 1.05% | ta | Tamil | 79M | 0.09% |
| ru | Russian | 145M | 9.16% | fa | Persian | 65M | 0.79% | jv | Javanese | 69M | 0.00% |
| ja | Japanese | 124M | 4.72% | pl | Polish | 38M | 1.69% | gu | Gujarati | 58M | 0.03% |
| ko | Korean | 81M | 0.84% | th | Thai | 38M | 0.64% | my | Burmese | 33M | 0.03% |
| de | German | 76M | 5.21% | uk | Ukrainian | 32M | 0.60% | pa | Punjabi | 32M | 0.01% |
| fr | French | 74M | 4.10% | ro | Romanian | 24M | 0.64% | tl | Tagalog | 28M | 0.02% |
| it | Italian | 63M | 2.33% | nl | Dutch | 23M | 1.57% | uz | Uzbek | 27M | 0.01% |
| | | | | el | Greek | 12M | 0.69% | az | Azerbaijani | 24M | 0.10% |
| | | | | bg | Bulgarian | 8M | 0.32% | ceb | Cebuano | 21M | 0.00% |
| | | | | hr | Croatian | 5.1M | 0.24% | sw | Swahili | 16M | 0.01% |
| | | | | sk | Slovak | 5M | 0.35% | km | Khmer | 16M | 0.02% |
| | | | | he | Hebrew | 5M | 0.27% | sq | Albanian | 7.5M | 0.05% |
| | | | | lt | Lithuanian | 2.8M | 0.18% | af | Afrikaans | 7M | 0.01% |
| | | | | lv | Latvian | 1.75M | 0.10% | no | Norwegian | 5.3M | 0.37% |
| | | | | et | Estonian | 1.1M | 0.14% | da | Danish | 5M | 0.36% |
| | | | | sv | Swedish | 10M | 0.63% | fi | Finnish | 5M | 0.41% |
| | | | | cs | Czech | 11M | 1.02% | is | Icelandic | 0.314M | 0.04% |
| | | | | hu | Hungarian | 13M | 0.49% | ga | Irish | — | 0.01% |
| | | | | sr | Serbian | 9M | 0.21% | ca | Catalan | 4M | 0.17% |
| | | | | sl | Slovenian | 2.1M | 0.13% | kk | Kazakh | 15M | 0.04% |
| | | | | | | | | kn | Kannada | 44M | 0.01% |
| | | | | | | | | ml | Malayalam | 38M | 0.02% |

### A.2    COMPARISON WITH OTHER WORK

Table 9 presents a comparison between MUBENCH, INCLUDE (Romanou et al., 2024), and BENCH-MAX (Huang et al., 2025). INCLUDE collects test questions from regional academic and professional certification exams, with a primary focus on local culture and knowledge. It supports 44 languages; however, the test samples are not aligned across languages and the number of samples per language varies significantly. BENCHMAX encompasses a broader range of task types to assess diverse model capabilities, including instruction following and code generation. Nevertheless, each task includes only a small number of samples. Although BENCHMAX is multilingual, it does not emphasize core multilingual capabilities such as natural language understanding and commonsense reasoning. In contrast, MUBENCH offers more comprehensive coverage in terms of language diversity, capability assessment, and sample volume. It aligns test samples across all supported languages and preserves fine-grained multilingual versions of each question—covering the instruction, question stem, and answer choices. This design enables high flexibility, facilitating the generation of variants tailored to different evaluation scenarios.

---

[3]https://en.wikipedia.org/wiki/List_of_languages_by_number_of_native_speakers

Table 9: Comparison of multilingual LLM benchmarks

| Benchmark | Supported Languages | Total Samples | Language Aligned | Variant Generation |
|-----------|---------------------|---------------|------------------|--------------------|
| MUBENCH | 61 | 3,921,751 | ✓ | ✓ |
| INCLUDE | 44 | 197,243 | ✗ | ✗ |
| BENCHMAX | 17 | 177,684 | ✓ | ✗ |

## A.3 DATASETS

MUBENCH focuses on core multilingual capabilities, including natural language understanding, commonsense reasoning, factual recall, knowledge-based question answering, academic and technical reasoning, and truthfulness. Therefore, we extend the most widely used English benchmarks for evaluating these capabilities to the multilingual setting. For each benchmark, we extend its test set to the multilingual setting and sample 50 examples from its training or validation set to serve as few-shot demonstrations.

**SNLI and MultiNLI** SNLI (Bowman et al., 2015) is a widely used dataset for evaluating natural language inference (NLI), where the task is to determine the logical relationship (entailment, contradiction, or neutral) between a given premise and hypothesis. It contains sentence pairs derived from image captions. MultiNLI (Williams et al., 2018) extends SNLI by including a broader range of genres, such as fiction, government, and telephone speech, making it a more diverse benchmark for evaluating models' generalization across different domains in NLI tasks. We use the mismatched validation set as the test set and matched validation set for few-shot demonstrations.

**StoryCloze** Story Cloze Test (Mostafazadeh et al., 2016) is a benchmark for evaluating a model's ability to understand narrative coherence and commonsense reasoning. Each example consists of a four-sentence story followed by two possible endings, and the task is to choose the more plausible ending. The dataset tests whether models can understand everyday events and make realistic predictions about what happens next in a story.

**WinoGrande** WinoGrande is a large-scale dataset comprising 44,000 problems, designed to evaluate commonsense reasoning in LLMs. Inspired by the original Winograd Schema Challenge (WSC) (Levesque et al.), WinoGrande addresses limitations of earlier datasets by increasing both the scale and difficulty of the tasks. Each problem presents a sentence with an ambiguous pronoun and two possible antecedents; the task is to determine the correct referent based on commonsense understanding. We use its validation set as the test samples.

**MMLU and MMLUPro** The MMLU (Hendrycks et al., 2021) dataset is a benchmark designed to assess language models' knowledge and reasoning across 57 subjects, including math, history, law, and medicine, using over 15,000 multiple-choice questions with four options each. MMLUPro (Wang et al., 2024) is an enhanced version that introduces more challenging questions, each with ten answer choices, making the task significantly harder and reducing the likelihood of guessing correctly. It is designed to better evaluate models' reasoning abilities and robustness across diverse prompts and domains.

**ARC** ARC (Clark et al., 2018) is a benchmark designed to evaluate the abilitie in advanced question answering. It comprises 7,787 multiple-choice science questions sourced from grade-school exams, divided into two subsets: the Easy Set and the Challenge Set. The Challenge Set includes questions that are difficult for simple retrieval or co-occurrence-based models.

**GPQA** GPQA (Rein et al., 2023) comprises 448 multiple-choice questions in biology, physics, and chemistry, crafted by domain experts to assess the reasoning abilities of both humans and LLMs, Designed to be exceptionally challenging.

**TruthfulQA** TruthfulQA (Lin et al., 2022a) is a benchmark designed to evaluate the truthfulness of language models in generating answers to diverse questions. The benchmark includes 817 questions covering 38 categories and targets "imitative falsehoods," which are false answers that resemble

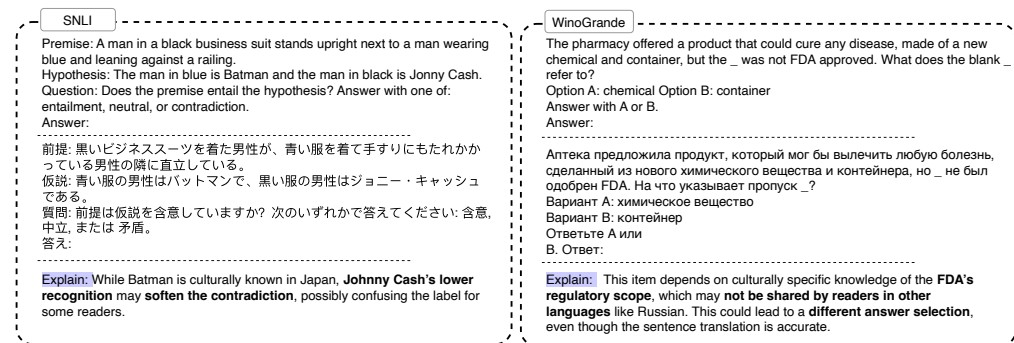

Figure 6: Cultural or background sensitive samples.

common misconceptions found in the models' training data. The goal is to assess the likelihood of models producing false or deceptive information without task-specific fine-tuning. We expand its validation set as our test set.

**BMLAMA** BMLAMA (Qi et al., 2023) is designed to evaluate the cross-lingual consistency of factual knowledge in multilingual LLMs. The test questions in this benchmark are aligned across all languages. We expand the 17-language version, BMLAMA-17, which contains 6,792 samples per language. However, upon inspection, we found numerous issues in BMLAMA-17, including inconsistencies among answer choices across different language versions. Therefore, we re-extended the dataset from its English version to 61 languages. MuBench does not include the original non-English samples from BMLAMA.

**HellaSwag** HellaSwag (Zellers et al., 2019) is a sentence completion task designed to test commonsense reasoning. Each example provides a short context followed by four possible sentence endings, and the model must choose the most plausible one. The incorrect options are crafted to be grammatically and stylistically similar, making the task challenging and requiring more than just surface-level understanding.

## A.4 CULTURAL SENSITIVITY

Analyzing the culturally sensitive samples reveals that, although our prompt instructed GPT-4o to flag only cases where cultural differences clearly influence the correct answer, the model adopted a more conservative criterion. It frequently identified content involving religion, region-specific knowledge, and niche cultural references as culturally sensitive. Given that the original datasets were created in English and contain numerous Western—particularly U.S.-centric—cultural assumptions, removing such samples helps mitigate cultural bias and supports a fairer, more balanced evaluation of LLMs across languages. Figure 6 illustrates two examples of culturally sensitive cases.

Table 10 presents a comparison between human experts and GPT-4o in labeling samples for cultural adaptability. Human experts identified significantly fewer culturally sensitive samples than GPT-4o. However, when we separately examine the human annotations for samples that GPT-4o labeled as sensitive and non-sensitive, we find that samples flagged as sensitive by GPT-4o are much more likely to be marked as sensitive by human experts as well.

Case analysis reveals that GPT-4o tends to flag samples involving niche cultural references tied to specific regions, religious topics, or similar themes. More specifically, because these datasets originate in English, they contain a substantial number of samples with a Western-centric perspective. While such content may not directly hinder the ability to answer the original questions, it implicitly assumes that LLMs respond from a Western cultural background. Using such samples to evaluate multilingual models may introduce or amplify regional and cultural biases in the development of LLMs.

As a result, we excluded samples labeled as culturally sensitive by GPT-4o from the final dataset. The impact of these samples on model behavior will be further investigated in future work.

Table 10: Per-language Cultural Sensitivity Agreement between GPT and Human Annotators

| Language | n | GPT True Count | Human True \| GPT=True | Human True \| GPT=False |
|---|---|---|---|---|
| id | 2452 | 1193 | 0.023 | 0.004 |
| de | 2155 | 980 | 0.042 | 0.018 |
| ms | 2159 | 958 | 0.313 | 0.013 |
| fr | 2008 | 927 | 0.033 | 0.026 |
| tr | 1893 | 901 | 0.069 | 0.011 |
| ru | 1830 | 856 | 0.105 | 0.017 |
| ja | 1850 | 848 | 0.134 | 0.046 |
| it | 2128 | 839 | 0.156 | 0.061 |
| zh | 1849 | 706 | 0.540 | 0.160 |
| es | 1745 | 669 | 0.027 | 0.005 |
| th | 1917 | 669 | 0.039 | 0.021 |
| nl | 1667 | 653 | 0.230 | 0.229 |
| pt | 1807 | 555 | 0.040 | 0.013 |
| ko | 1762 | 485 | 0.165 | 0.046 |
| vi | 1619 | 450 | 0.013 | 0.006 |
| tl | 1640 | 325 | 0.332 | 0.077 |

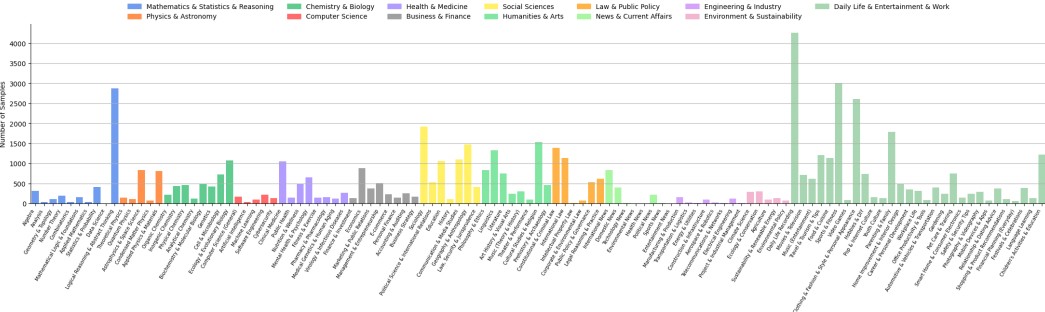

Figure 7: Sample content classification.

## A.5 DIVERSITY

Figure 7 presents the distribution of all MUBENCH samples based on the two-level classification scheme. MUBENCH demonstrates substantial diversity, encompassing a broad spectrum of academic disciplines and everyday topics. Daily life scenarios constitute a significant portion of the dataset, largely contributed by sources such as SNLI, MultiNLI, StoryCloze, and Winogrande. This diversity in real-world content is crucial for assessing the semantic understanding capabilities of LLMs across multiple languages.

## A.6 QUALITY CONTROL

We recruited human annotators who hold at least a college degree, possess C1-level English proficiency or equivalent certification, and are native speakers of the languages they were assigned to evaluate.

We first extract all samples labeled as culturally sensitive by GPT-4o from SNLI, MNLI, WinoGrande, HellaSwag, BMLAMA, ARCEasy, ARCChallenge, StoryCloze, and MMLU. Then, we perform sampling based on semantic consistency scores and language purity scores estimated by GPT-4o during the translation process, aiming to ensure that there are at least 30 samples for each score level whenever possible. Additionally, we include samples extracted from OpenAI's MMMLU. All selected samples are then submitted to human experts for evaluation of semantic consistency, purity, and cultural sensitivity, using the same rubrics as those employed by GPT-4o. Notably, when asking human experts to evaluate semantic consistency, we directly provide the original and translated versions without performing back-translation.

Table 11 presents the average scores given by human experts and GPT-4o for each dataset. It can be observed that GPT-4o generally rates the translations more strictly than human evaluators. Across the datasets, the expert scores do not show significant variation, indicating that the translation quality is consistently high regardless of the dataset content.

Table 11: Comparison of Human and GPT Consistency and Purity Scores across Datasets

| Dataset | Samples | Semantic Consistency | | Translation Purity | |
|---|---|---|---|---|---|
| | | Human | GPT | Human | GPT |
| MNLI | 3757 | 4.6577 | 3.4195 | 4.5885 | 3.4482 |
| SNLI | 3623 | 4.6953 | 3.7248 | 4.6539 | 3.8184 |
| ARCEasy | 2561 | 4.8684 | 4.0016 | 4.8134 | 4.3811 |
| ARCChallenge | 2161 | 4.8903 | 4.1731 | 4.8066 | 4.3734 |
| WinoGrande | 2643 | 4.7499 | 4.0851 | 4.7662 | 3.5634 |
| BMLAMA | 1499 | 4.8953 | 4.3062 | 4.5264 | 3.5911 |
| Hellaswag | 3668 | 4.7001 | 4.2435 | 4.4959 | 3.4602 |
| StoryCloze | 2389 | 4.7401 | 4.3713 | 4.6756 | 3.7874 |
| MMLU | 8180 | 4.7605 | 4.4189 | 4.6632 | 3.8302 |

Figure 9 exhibits the distributions of semantic consistency and purity scores in each language rated by GPT-4o.

## B  LINGUISTIC TYPOLOGY INFLUENCES ON LANGUAGE CONSISTENCY

We report GPT-4o's MLC scores on our 61-language MMLU in Table 12. The mean and standard deviation of intra-group MLC scores are presented using three language typology classifications: **language family**, **morphological type**, and **word order**. Only groups with more than two languages are included. We observe that language families generally exhibit high intra-group MLC, while groups based on word order or morphology show lower consistency.

Table 12: Intra-group MLC scores by Language Family, Morphological Type, and Word Order

| Language Family | | | Morphological Type | | | Word Order Type | | |
|---|---|---|---|---|---|---|---|---|
| Family | Mean | Std | Type | Mean | Std | Order | Mean | Std |
| Austronesian | 83.04 | 1.49 | Agglutinative | 66.48 | 20.90 | Flexible/Mixed | 62.38 | 26.04 |
| Dravidian | 74.54 | 2.80 | Analytic | 75.84 | 10.14 | SOV | 69.01 | 18.67 |
| Germanic | 55.02 | 25.61 | Fusional | 68.72 | 22.47 | SVO | 73.62 | 17.63 |
| Indo-Aryan | 80.67 | 2.47 | | | | | | |
| Romance | 84.80 | 3.89 | | | | | | |
| Slavic | 85.05 | 1.90 | | | | | | |
| Turkic | 49.25 | 28.43 | | | | | | |
| Uralic | 82.54 | 2.26 | | | | | | |

## C  PARALLEL CORPORA IMPACT STUDY

Consistency and accuracy together provide a holistic view of a model's multilingual capabilities, revealing both performance and the extent of cross-lingual transfer. Enhancing such transfer remains a key open challenge. While parallel corpora are commonly used to improve cross-lingual generalization, their exact contribution is not well understood. To explore this, we conduct experiments examining how incorporating parallel data under different language ratio settings affects model performance across languages.

**Experimental Setup**  We pretrain 1.2B-parameter LLaMA-2 models on Chinese-English and Arabic-English corpora—two linguistically distant, high-resource languages—cleaned from CommonCrawl. Training is done under two data distributions: (1) equal Chinese-English, and (2) a 1:9 ratio of Chinese-to-English / Arabic-to-English, with total tokens fixed at 500B. To assess the impact of parallel data, we translate English into Chinese and Arabic and filter with COMET (Rei

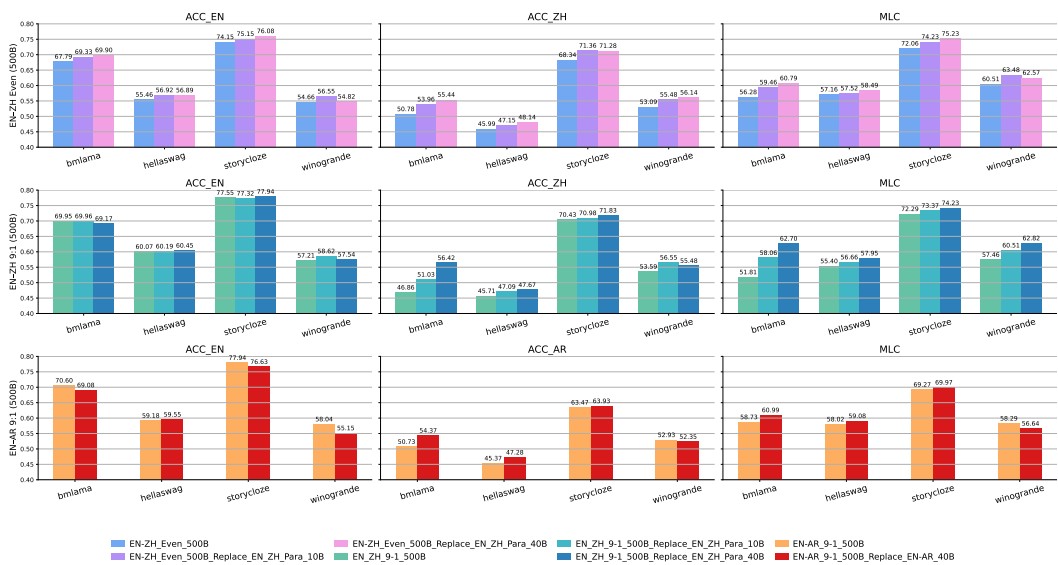

Figure 8: Impact of parallel corpus proportion on language proficiency.

et al., 2020), keeping only pairs with scores above 0.8, yielding 10B and 40B tokens of parallel data. In each setting, we replace 10B or 40B monolingual tokens—removing equal parts from both languages—to maintain the total token count.

**Result** Figure 8 shows the performance of 1.2B-parameter models on natural language understanding and factual knowledge tasks. Even under equal data distribution, English consistently outperforms Chinese, reflecting its dominance in global data availability. For the Chinese-English setting, introducing parallel corpora improves overall performance across both data settings, with gains primarily observed in Chinese. This, along with increased consistency, suggests that some English capabilities are effectively transferred to Chinese. Conversely, modest improvements in English performance under the equal distribution setting indicate reciprocal benefits from Chinese. Since both languages are limited to 250B tokens, excessive parallel data—despite its transfer benefits—can reduce overall information diversity due to redundancy. This is evident as the performance gain from 40B tokens of parallel data is marginal compared to 10B. In the 90% English setting, however, even a small amount of parallel data significantly boosts Chinese performance, matching that of a model trained on 250B Chinese tokens. Yet, diminishing returns and even regression (e.g., on WinoGrande) with 40B tokens, which is also the case in Arabic, highlight potential drawbacks of overusing parallel data. These results reveal a trade-off between preserving information diversity and enhancing cross-lingual transfer, shaped by data ratios and parallel corpus size. They also showcase MUBENCH's value in probing multilingual dynamics and guiding future LLM development.

# D IMPLEMENTATION DETAILS

## D.1 EVALUATION

All evaluations of open-source models were conducted on a single 8×H100 GPU cluster node. The evaluation code was based on the Hugging Face Transformers[4] library, and for models larger than 20B parameters, we used vLLM[5] for inference.

## D.2 PARALLEL CORPORA EXPERIMENT

**Training Data** We process Common Crawl snapshots with deduplication and heuristic filtering pipelines inspired by SlimPajama (Sli) and FineWeb-Edu (Penedo et al., 2024).

---

[4] https://huggingface.co/docs/transformers
[5] https://github.com/vllm-project/vllm

**Model Architecture** We adopt a transformer architecture based on the LLaMA-2 model, scaled to approximately 1.2 billion parameters. All models are initialized randomly prior to pretraining. Table 13 provides the full configuration details and training hyperparameters. To support training, we construct a custom Byte-Pair Encoding (BPE) tokenizer using the BBPE algorithm, resulting in a vocabulary of 250,000 tokens. The primary experiments are run on 64 NVIDIA H100 GPUs, with each experiment taking roughly 50 hours on average.

| Model Configuration | Value |
|---|---|
| Number of attention heads | 16 |
| Number of layers | 24 |
| Hidden size | 2048 |
| Intermediate layer dimension | 5504 |
| Maximum position embeddings | 4096 |
| Layer normalization epsilon | $1 \times 10^{-5}$ |
| **Training Hyperparameters** | **Value** |
| Batch size | 3072 |
| Sequence length | 4096 |
| Optimizer | AdamW |
| Learning rate | $4.3 \times 10^{-4}$ |
| Learning rate schedule | Cosine decay to 10% of initial value |
| Training steps | Varied based on total token budget |
| Precision | bfloat16 (mixed-precision training) |

Table 13: Model configuration and training hyperparameters used for LLM pretraining.

## D.3 PROMPT DESIGN

The following contains the prompts used during the construction of MuBench, including the main stages content classification, translation, semantic consistency evaluation, translation purity assessment, and cultural sensitivity check. For semantic consistency evaluation, first the back translation is involved and then the scoring follows.

---

**Translation Prompt**

Please translate the entire text, into {target language}.

Translate **all content**, including prompt indicators (e.g., Premise, Hypothesis, Question, Choice, Option, Answer, header, title, step, substeps, etc.), partial phrases, and any other English words or phrases. **Do not** leave any part untranslated.
**Strictly preserve:**

- All original HTML tags (such as `<p>`, ``, `<s1>`, etc.) and their structure
- All special symbols and placeholders, especially underscores _ which indicate missing words or pronouns
- Option labels such as A, B, C, D (used in multiple-choice questions) must remain unchanged
- All line breaks, punctuation, and formatting

The translation must **not** reveal or imply the correct answer. **Do not** modify the wording in a way that would make one choice obviously correct or invalid.
The question must remain **valid, challenging, and unbiased**, as in the original English.
The translation must also be natural, fluent, and contextually appropriate, as if written by a native speaker.
**Do not** add any explanation, annotation, or commentary.

**Text:**

{original text}

---

**Back Translation Prompt**

Please translate the text back into English.
Strictly preserve all original HTML tags (such as `<p></p>`), formatting, punctuation, line breaks, and structure.
Do **not** answer any questions or interpret the meaning — just provide a **faithful translation** of the text.
Do **not** add any explanation or commentary.

**Text:**
{translated text}

---

**Semantic Consistency Scoring Prompt**

You will be given two English texts: an original and a rewritten version.
Score the rewritten version's consistency with the original on a scale of 1 to 5, based on these criteria:
5 points: Completely consistent — the rewritten version conveys exactly the same meaning as the original.
4 points: Mostly consistent — only very minor wording changes with no effect on understanding.
3 points: Generally consistent — some differences that might slightly confuse.
2 points: Significant differences — clear changes that can affect the answer.
1 point: Completely inconsistent — the meaning has fundamentally changed.

**Original Text:**
{original text}

**Rewritten Text:** {back translated text}

Only output a single digit between 1 and 5.

---

**Cultural Sensitivity Judgment Prompt**

Please determine whether the following question contains cultural context or background that would definitely cause the meaning or correct answer to change when translated into {target language}.
Only respond with "Yes" if there is a clear cultural difference that would lead to a different interpretation or answer in the target language. If you are not sure or if no such difference exists, respond with "No". Do not explain your reasoning.

**Text:**
{original text}

**Translated Text:**
{translated text}

---

**Category Prompt**

Please choose the most relevant category for this text, focusing on the content and scenario described in the question stem or the main body of the text, rather than the question type or answer format.
Categories: {categories}. Only output one of these categories without any explanation, even if the question type might be misleading.

**Text:**
{text}

---

**Language Purity Scoring Prompt**

Evaluate the language purity of the text, based on how fully it is written in {Target Language}.

Give a score from 1 to 5, where:

5 — The text is written entirely in {Target Language}, with **no English words at all**, not even one.

4 — The text is mostly in {Target Language}, but includes a few English loanwords, brand names, or transliterations that are commonly accepted.

3 — The text contains some English words, names, or abbreviations that are not necessary and could have been translated.

2 — The text mixes {Target Language} with many English terms that break the language flow and reduce clarity.

1 — The text contains a large amount of English or appears heavily code-mixed, making it hard to identify {Target Language} as the dominant language.

Ignore option labels such as A, B, C, D — they are not considered part of the language and should not affect the score.

Only reply with a number from 1 to 5. Do not include any explanation or reasoning.

**Text to evaluate:**
{translated text}

---

## E  FULL RESULTS

MuBench includes datasets of varying difficulty levels. Some test sets are particularly challenging for base models. Due to space limitations, we only present the results on key datasets in the main text and omit those test sets that are excessively difficult for base models such as MMLUPro and GPQA.

Table 14 presents the full results on all datasets of MUBENCH. Table 15 shows the full results of multilingual consistency on all datasets of MUBENCH.

## F  COST ESTIMATION

We used GPT-4o-2024-05-13 for all translations, with a total cost of approximately $57,038. In addition, using GPT-4o-2024-05-13 for evaluation across all languages incurred a total cost of $6,441. Evaluating all other open-source models required approximately 8,064 H100 GPU hours. The cost of human expert evaluations was around $31,212. Annotators were paid at an hourly rate of $16, with a maximum of 8 working hours per day.

Table 14: Performance of LLMs on MuBench. The values in parentheses indicate the score differences relative to English performance.

| | SNLI | MultiNLI | StoryCloze | WinoGrande | BMLAMA | MMLU | MMLUPro | HellaSwag | GPQA | ARCEasy | ARCChallenge | TruthfulQA |
|---|---|---|---|---|---|---|---|---|---|---|---|---|
| **Proprietary Model** | | | | | | | | | | | | |
| gpt-4o-2024-05-13 | 78.74 (-8.57) | 69.78 (-11.18) | 97.68 (-1.62) | 71.68 (-10.35) | 66.87 (-6.90) | 70.01 (-2.26) | 38.22 (-5.65) | 83.02 (-10.75) | 30.15 (+2.92) | 93.64 (-5.00) | 87.32 (-7.35) | 75.25 (-6.38) |
| **Model (1–4B)** | | | | | | | | | | | | |
| Qwen3-0.6B-Base | 41.21 (-25.96) | 38.45 (-30.53) | 56.05 (-15.78) | 50.67 (-6.20) | 27.17 (-32.19) | 26.88 (-5.38) | 9.12 (-2.11) | 31.01 (-21.29) | 22.16 (-0.83) | 29.75 (-19.25) | 24.62 (-8.89) | 28.60 (-2.86) |
| Qwen3-1.7B-Base | 54.36 (-31.13) | 56.33 (-24.75) | 59.71 (-17.84) | 50.99 (-6.30) | 31.89 (-28.45) | 28.13 (-7.30) | 10.41 (-4.46) | 35.68 (-28.29) | 23.06 (-1.49) | 33.46 (-23.00) | 26.88 (-9.80) | 29.83 (-1.80) |
| Qwen3-4B-Base | 72.06 (-10.42) | 69.26 (-4.47) | 64.16 (-17.19) | 53.27 (-10.04) | 37.82 (-26.87) | 30.18 (-8.38) | 12.81 (-5.82) | 42.52 (-29.57) | 22.19 (-1.25) | 37.55 (-19.51) | 30.09 (-9.43) | 31.25 (-0.89) |
| Qwen2.5-0.5B | 35.39 (-28.03) | 35.10 (-25.94) | 54.26 (-17.10) | 50.39 (-3.44) | 26.42 (-39.55) | 26.27 (-4.85) | 8.71 (-1.46) | 29.42 (-20.54) | 21.36 (-0.52) | 28.06 (-21.83) | 23.67 (-7.34) | 25.45 (-4.14) |
| Sailor2-1B | 34.30 (-20.58) | 34.56 (+2.06) | 54.82 (-18.32) | 49.98 (-5.50) | 28.37 (-37.95) | 26.22 (-3.45) | 8.57 (-0.11) | 29.88 (-20.30) | 21.94 (+0.06) | 28.83 (-18.18) | 23.51 (-5.79) | 26.04 (-2.36) |
| Qwen2.5-1.5B | 46.19 (-41.99) | 46.11 (-29.98) | 56.17 (-24.63) | 50.48 (-10.94) | 31.91 (-37.04) | 27.19 (-7.73) | 9.34 (-3.36) | 31.64 (-33.95) | 21.80 (-2.53) | 29.51 (-24.67) | 24.62 (-12.92) | 27.37 (-3.92) |
| gemma-3-1b-pt | 32.89 (+0.75) | 32.66 (+0.22) | 56.91 (-10.74) | 51.62 (-5.76) | 41.71 (-27.31) | 26.62 (-1.29) | 10.36 (-0.71) | 31.11 (-13.02) | 22.41 (-1.25) | 28.94 (-7.77) | 24.84 (-2.05) | 29.83 (-0.78) |
| gemma-3-4b-pt | 43.20 (-15.41) | 42.48 (-5.82) | 58.31 (-9.65) | 56.01 (-11.43) | 52.57 (-17.96) | 26.70 (-1.40) | 9.99 (-0.37) | 34.31 (-16.81) | 22.63 (-2.15) | 29.26 (-10.08) | 24.47 (-2.94) | 27.35 (+0.31) |
| gemma-2-2b | 36.43 (+0.63) | 34.51 (-12.74) | 63.98 (-18.91) | 52.53 (-11.94) | 40.48 (-30.73) | 28.05 (-6.27) | 11.27 (-4.30) | 40.29 (-30.46) | 22.42 (-1.46) | 33.45 (-16.53) | 27.36 (-8.81) | 30.74 (-0.89) |
| **Model (7–20B)** | | | | | | | | | | | | |
| Qwen3-8B-Base | 80.12 (-6.73) | 76.16 (-6.56) | 67.87 (-16.42) | 55.41 (-12.13) | 47.44 (-24.70) | 31.47 (-8.14) | 14.09 (-6.00) | 47.72 (-28.02) | 24.30 (-2.49) | 40.51 (-17.90) | 31.73 (-8.13) | 32.42 (-0.23) |
| Qwen3-14B-Base | 84.20 (-3.59) | 81.63 (-0.92) | 71.14 (-13.61) | 57.67 (-15.04) | 51.72 (-21.14) | 32.61 (-8.22) | 15.74 (-6.15) | 52.86 (-25.90) | 26.08 (-2.04) | 42.75 (-15.41) | 33.71 (-5.98) | 33.54 (-2.68) |
| Qwen2.5-7B | 68.28 (-21.13) | 67.23 (-18.14) | 61.88 (-22.02) | 51.68 (-14.68) | 36.02 (-28.39) | 29.77 (-9.56) | 11.76 (-5.27) | 39.52 (-36.92) | 22.91 (-1.87) | 35.49 (-24.49) | 28.14 (-11.98) | 28.96 (-4.88) |
| Sailor2-8B | 52.81 (-27.29) | 54.66 (-25.99) | 61.96 (-20.62) | 52.59 (-11.96) | 40.26 (-30.47) | 28.25 (-7.76) | 10.09 (-3.81) | 38.44 (-34.76) | 22.77 (-0.22) | 34.11 (-22.44) | 26.62 (-11.01) | 27.56 (-1.52) |
| Babel-9B | 68.26 (-21.89) | 66.38 (-22.27) | 61.89 (-21.48) | 53.29 (-14.72) | 42.73 (-29.34) | 29.15 (-9.30) | 11.76 (-5.34) | 40.57 (-34.25) | 22.84 (-1.04) | 34.25 (-27.73) | 27.64 (-13.08) | 28.26 (-1.84) |
| Qwen2.5-14B | 76.04 (-10.95) | 74.24 (-11.83) | 66.50 (-19.26) | 50.19 (-11.89) | 23.68 (-31.04) | 31.64 (-9.70) | 13.92 (-5.82) | 45.62 (-35.09) | 24.03 (-1.19) | 39.05 (-20.59) | 31.20 (-11.07) | 31.13 (-0.84) |
| Sailor2-20B | 75.26 (-15.83) | 73.36 (-16.07) | 67.41 (-18.50) | 56.30 (-18.64) | 48.11 (-25.13) | 30.61 (-8.94) | 12.71 (-5.62) | 46.74 (-32.83) | 24.10 (+0.22) | 38.14 (-20.95) | 30.36 (-10.71) | 30.02 (-4.33) |
| gemma-3-12b-pt | 51.85 (-15.21) | 37.08 (-4.45) | 55.42 (-4.02) | 61.40 (-11.56) | 59.61 (-12.17) | 26.27 (-0.87) | 10.13 (+0.67) | 30.50 (-4.01) | 23.26 (+0.05) | 28.27 (-3.40) | 24.23 (-1.21) | 26.22 (-0.82) |
| gemma-2-9b | 69.92 (-5.87) | 65.10 (-12.05) | 73.40 (-12.28) | 57.98 (-13.83) | 53.59 (-18.12) | 31.64 (-7.18) | 14.87 (-4.44) | 55.66 (-22.19) | 24.20 (-1.92) | 41.75 (-13.91) | 33.27 (-7.11) | 32.25 (+0.28) |
| **Model (≥20B)** | | | | | | | | | | | | |
| Qwen2.5-32B | 81.67 (-9.51) | 80.36 (-7.61) | 68.19 (-18.57) | 56.95 (-17.91) | 48.84 (-23.45) | 33.30 (-8.51) | 16.09 (-5.47) | 49.43 (-32.07) | 24.15 (-3.53) | 41.51 (-17.96) | 33.12 (-10.95) | 31.85 (-3.69) |
| Qwen2.5-72B | 84.63 (-6.63) | 84.48 (-5.53) | 71.89 (-15.42) | 59.17 (-18.82) | 52.87 (-19.79) | 36.25 (-7.59) | 18.56 (-4.57) | 54.99 (-28.77) | 25.86 (-1.15) | 46.73 (-15.50) | 36.40 (-9.13) | 34.53 (-3.23) |
| Babel-83B | 85.68 (-5.86) | 85.29 (-5.04) | 71.40 (-15.83) | 58.52 (-18.89) | 52.46 (-20.91) | 34.75 (-8.19) | 17.52 (-5.06) | 54.65 (-28.33) | 26.06 (-2.06) | 43.08 (-18.51) | 34.47 (-8.06) | 32.88 (-4.36) |
| gemma-3-27b-pt | 81.71 (-5.27) | 77.12 (-8.60) | 79.06 (-8.48) | 63.49 (-13.34) | 61.74 (-10.48) | 36.46 (-4.84) | 19.19 (-4.16) | 66.09 (-14.28) | 27.55 (-1.47) | 48.18 (-7.01) | 37.99 (-3.59) | 32.19 (-0.46) |
| gemma-2-27b | 79.28 (-8.92) | 75.38 (-8.58) | 77.21 (-10.17) | 60.78 (-15.81) | 56.09 (-14.85) | 34.09 (-6.76) | 17.41 (-4.23) | 62.08 (-20.02) | 26.40 (-1.72) | 44.23 (-9.48) | 35.70 (-3.90) | 32.40 (-2.46) |

Table 15: Consistency across languages. 'All' refers to the average consistency across all language pairs, while 'vs. EN' indicates the average consistency between each language and English.

| | SNLI | | MultiNLI | | StoryCloze | | WinoGrande | | BMLAMA | | MMLU | | MMLUPro | | HellaSwag | | GPQA | | ARCEasy | | ARCChallenge | | TruthfulQA | |
|---|---|---|---|---|---|---|---|---|---|---|---|---|---|---|---|---|---|---|---|---|---|---|---|---|
| | All | vs. EN | All | vs. EN | All | vs. EN | All | vs. EN | All | vs. EN | All | vs. EN | All | vs. EN | All | vs. EN | All | vs. EN | All | vs. EN | All | vs. EN | All | vs. EN |
| **Proprietary Model** | | | | | | | | | | | | | | | | | | | | | | | | |
| gpt-4o-2024-05-13 | 78.37 | 83.63 | 74.60 | 79.25 | 96.71 | 98.06 | 74.93 | 78.54 | 66.21 | 74.67 | 68.42 | 69.71 | 42.46 | 47.07 | 83.37 | 86.95 | 47.46 | 43.93 | 90.34 | 94.28 | 84.52 | 89.24 | 80.62 | 83.63 |
| **Model (1–4B)** | | | | | | | | | | | | | | | | | | | | | | | | |
| Qwen3-0.6B-Base | 42.32 | 48.53 | 49.51 | 51.04 | 64.15 | 62.68 | 55.10 | 57.79 | 29.64 | 35.36 | 49.22 | 48.98 | 44.84 | 42.07 | 49.68 | 45.60 | 64.00 | 63.52 | 39.42 | 40.44 | 40.94 | 41.26 | 55.71 | 51.53 |
| Qwen3-1.7B-Base | 53.28 | 61.02 | 56.72 | 62.92 | 65.67 | 67.12 | 55.84 | 54.96 | 33.92 | 42.06 | 49.82 | 50.21 | 44.14 | 42.48 | 50.91 | 48.79 | 64.00 | 64.70 | 41.45 | 43.91 | 42.66 | 43.48 | 56.39 | 54.98 |
| Qwen3-4B-Base | 71.09 | 75.54 | 70.39 | 70.99 | 67.89 | 69.64 | 57.28 | 60.18 | 36.24 | 44.74 | 51.08 | 52.36 | 44.42 | 43.41 | 53.48 | 54.04 | 64.16 | 65.36 | 43.54 | 46.92 | 44.13 | 45.65 | 56.40 | 54.87 |
| Qwen2.5-0.5B | 41.39 | 32.35 | 42.98 | 48.39 | 62.45 | 60.86 | 54.09 | 54.11 | 27.93 | 34.21 | 47.67 | 45.94 | 45.06 | 40.15 | 47.63 | 42.30 | 62.83 | 60.68 | 37.17 | 37.19 | 39.40 | 38.03 | 54.06 | 51.49 |
| Sailor2-1B | 49.24 | 15.33 | 58.48 | 71.72 | 62.71 | 61.63 | 54.75 | 55.64 | 28.96 | 36.57 | 48.64 | 48.28 | 46.54 | 43.36 | 47.89 | 43.45 | 63.88 | 62.98 | 38.56 | 39.51 | 40.45 | 40.46 | 53.76 | 52.19 |
| Qwen2.5-1.5B | 43.09 | 52.32 | 45.29 | 55.07 | 63.38 | 62.97 | 54.18 | 55.19 | 32.64 | 40.60 | 48.31 | 47.52 | 43.53 | 39.96 | 48.21 | 43.50 | 63.44 | 61.53 | 38.52 | 39.92 | 39.87 | 38.64 | 54.14 | 51.67 |
| gemma-3-1b-pt | 42.90 | 44.96 | 86.21 | 92.58 | 65.71 | 66.21 | 55.84 | 58.85 | 40.64 | 51.17 | 52.79 | 54.52 | 48.13 | 48.67 | 50.64 | 50.74 | 65.85 | 67.30 | 40.81 | 43.46 | 42.78 | 43.76 | 52.52 | 53.49 |
| gemma-3-4b-pt | 43.49 | 48.92 | 42.04 | 44.77 | 66.11 | 66.30 | 59.46 | 62.38 | 51.35 | 61.04 | 50.89 | 53.02 | 45.52 | 45.80 | 51.50 | 51.40 | 64.08 | 64.18 | 39.47 | 42.46 | 41.23 | 42.81 | 51.49 | 52.83 |
| gemma-2-2b | 35.45 | 43.90 | 41.50 | 29.26 | 67.46 | 70.27 | 55.75 | 58.26 | 39.24 | 49.81 | 53.82 | 54.36 | 48.60 | 47.53 | 51.53 | 52.15 | 67.84 | 68.35 | 43.21 | 46.23 | 44.09 | 45.98 | 56.68 | 56.45 |
| **Model (7–20B)** | | | | | | | | | | | | | | | | | | | | | | | | |
| Qwen3-8B-Base | 78.37 | 81.58 | 74.47 | 78.24 | 69.90 | 72.52 | 58.79 | 62.14 | 45.48 | 55.63 | 51.39 | 52.52 | 44.59 | 43.79 | 56.02 | 58.20 | 64.79 | 66.12 | 45.23 | 48.91 | 45.21 | 46.85 | 57.40 | 56.00 |
| Qwen3-14B-Base | 83.45 | 83.85 | 80.76 | 79.75 | 72.01 | 74.85 | 60.12 | 63.72 | 49.80 | 59.66 | 52.59 | 54.06 | 45.02 | 44.74 | 58.75 | 61.95 | 64.85 | 66.84 | 46.26 | 49.78 | 46.01 | 48.45 | 58.61 | 58.76 |
| Qwen2.5-7B | 65.59 | 74.58 | 65.37 | 74.53 | 66.10 | 68.76 | 54.96 | 56.65 | 34.49 | 42.58 | 49.28 | 50.29 | 42.92 | 42.04 | 50.72 | 51.05 | 61.89 | 62.62 | 41.29 | 45.31 | 41.33 | 43.29 | 55.19 | 54.90 |
| Sailor2-8B | 53.75 | 62.30 | 49.86 | 60.37 | 66.55 | 68.44 | 56.51 | 58.37 | 38.36 | 48.17 | 50.40 | 50.93 | 44.98 | 43.13 | 51.13 | 50.40 | 64.89 | 64.63 | 42.04 | 44.87 | 42.07 | 43.57 | 54.14 | 53.91 |
| Babel-9B | 62.61 | 72.29 | 58.75 | 69.49 | 65.08 | 68.86 | 57.03 | 59.66 | 40.97 | 51.46 | 46.99 | 49.04 | 40.81 | 40.83 | 50.50 | 52.44 | 61.82 | 63.79 | 39.53 | 44.21 | 39.66 | 42.06 | 47.47 | 50.70 |
| Qwen2.5-14B | 74.94 | 80.05 | 74.97 | 79.11 | 68.59 | 72.11 | 57.45 | 56.17 | 26.19 | 31.21 | 50.08 | 51.71 | 43.35 | 42.79 | 53.83 | 56.40 | 62.96 | 64.29 | 43.20 | 47.79 | 42.89 | 45.41 | 56.55 | 55.89 |
| gemma-2-9b | 75.07 | 80.40 | 70.00 | 74.91 | 73.61 | 77.25 | 60.18 | 64.31 | 51.62 | 61.12 | 55.87 | 57.51 | 50.12 | 49.77 | 60.41 | 64.38 | 71.00 | 73.30 | 47.12 | 51.71 | 47.02 | 49.58 | 57.85 | 57.82 |
| **Model (≥20B)** | | | | | | | | | | | | | | | | | | | | | | | | |
| Qwen2.5-32B | 81.18 | 85.94 | 80.83 | 84.48 | 69.48 | 72.74 | 58.74 | 61.98 | 46.54 | 56.12 | 50.91 | 52.75 | 43.07 | 43.12 | 55.57 | 59.19 | 61.88 | 63.80 | 44.21 | 48.69 | 43.74 | 47.11 | 57.01 | 56.19 |
| Qwen2.5-72B | 84.69 | 88.64 | 84.65 | 88.06 | 71.89 | 75.87 | 60.01 | 64.32 | 50.23 | 59.90 | 53.01 | 55.39 | 45.08 | 45.26 | 59.12 | 63.65 | 64.67 | 66.63 | 47.44 | 52.25 | 45.83 | 49.05 | 58.07 | 59.10 |
| Babel-83B | 86.30 | 89.37 | 85.20 | 88.34 | 71.83 | 75.46 | 60.00 | 64.44 | 50.17 | 59.73 | 52.70 | 55.09 | 45.46 | 45.64 | 59.17 | 63.59 | 65.66 | 66.39 | 46.24 | 50.90 | 45.59 | 48.59 | 55.89 | 57.83 |
| gemma-3-27b-pt | 82.68 | 86.53 | 77.43 | 82.09 | 78.16 | 81.36 | 64.50 | 69.68 | 61.02 | 68.10 | 58.66 | 61.91 | 53.24 | 54.88 | 68.55 | 72.63 | 73.72 | 74.46 | 52.16 | 55.87 | 51.07 | 54.29 | 60.85 | 61.94 |
| gemma-2-27b | 79.14 | 82.88 | 74.24 | 77.78 | 76.15 | 79.80 | 61.57 | 65.91 | 53.65 | 62.81 | 55.39 | 58.03 | 47.98 | 48.62 | 64.55 | 69.50 | 66.33 | 68.82 | 48.27 | 51.44 | 48.06 | 51.77 | 59.25 | 61.73 |

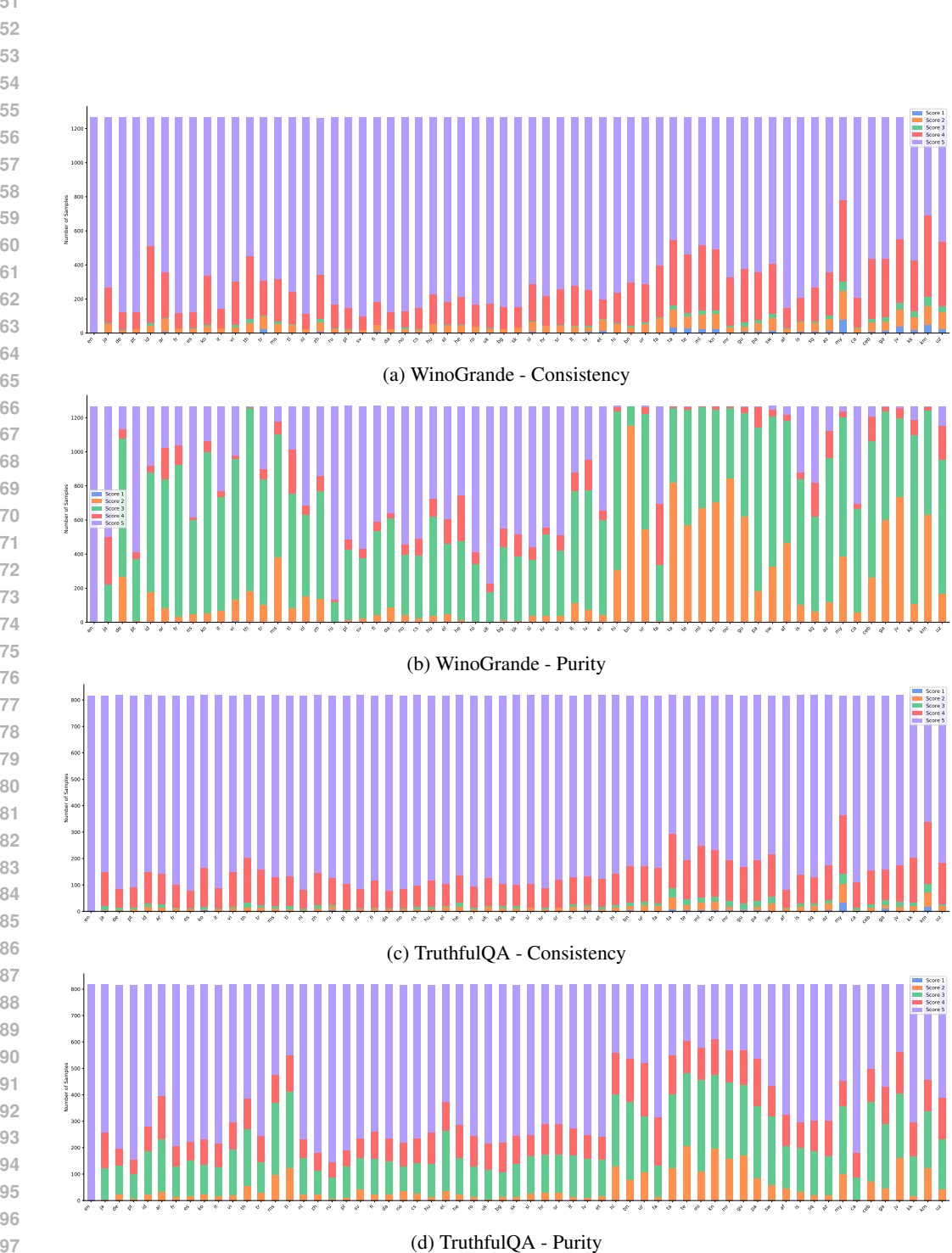

(a) WinoGrande - Consistency

(b) WinoGrande - Purity

(c) TruthfulQA - Consistency

(d) TruthfulQA - Purity

Figure 9: Consistency and purity distributions evaluated by GPT-4o (Part 1/6)

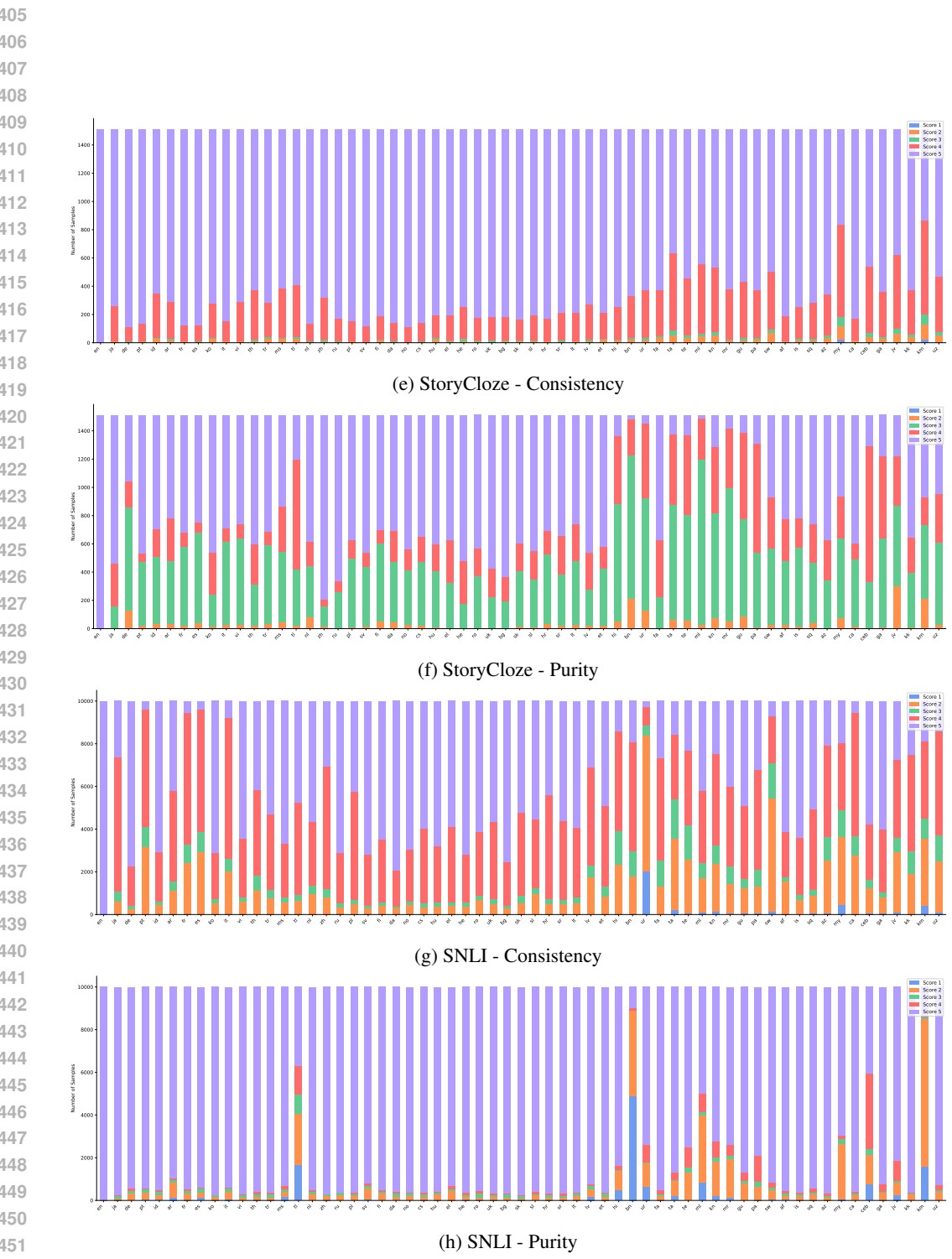

(e) StoryCloze - Consistency

(f) StoryCloze - Purity

(g) SNLI - Consistency

(h) SNLI - Purity

Figure 9: Consistency and purity distributions evaluated by GPT-4o (Part 2/6)

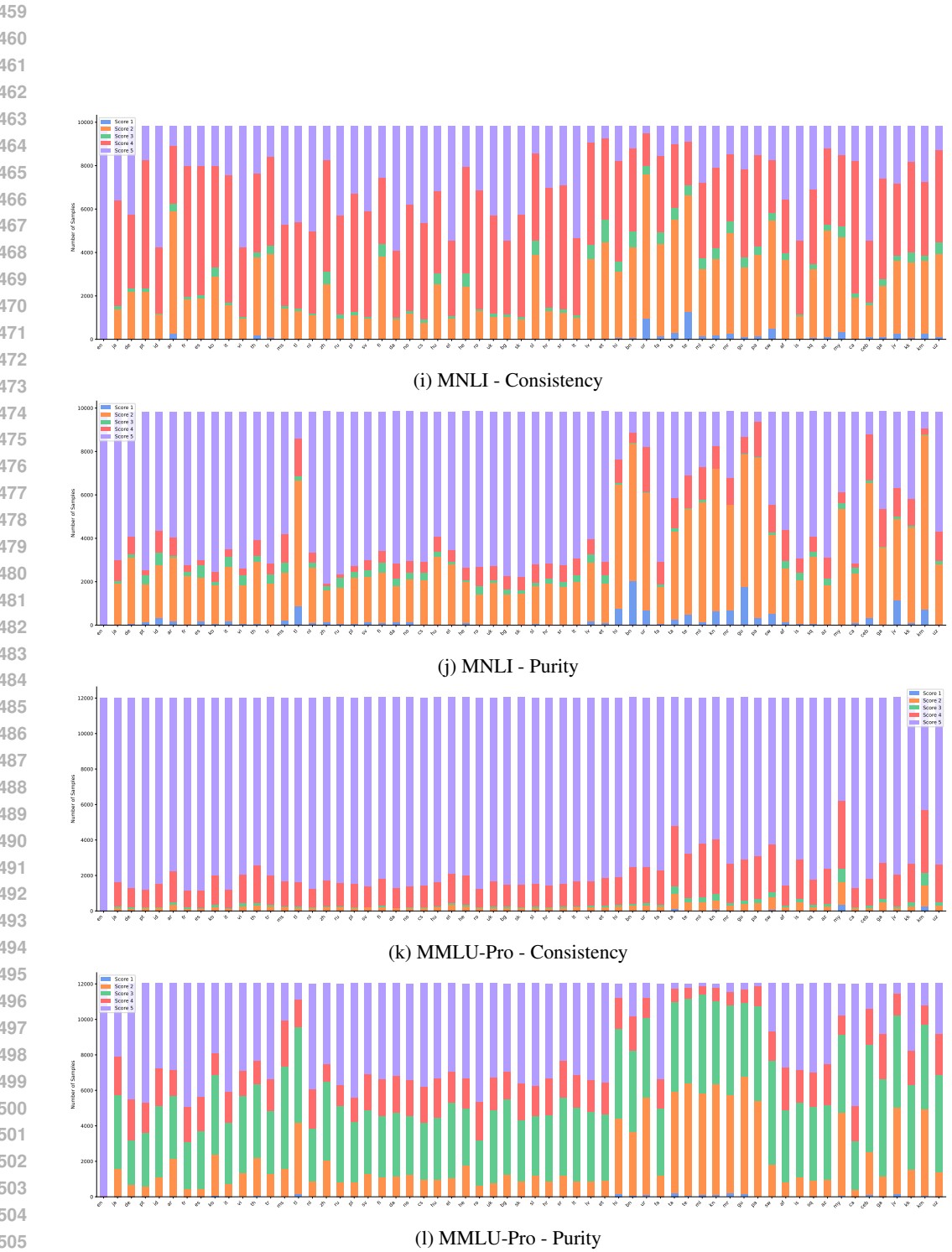

(i) MNLI - Consistency

(j) MNLI - Purity

(k) MMLU-Pro - Consistency

(l) MMLU-Pro - Purity

Figure 9: Consistency and purity distributions evaluated by GPT-4o (Part 3/6)

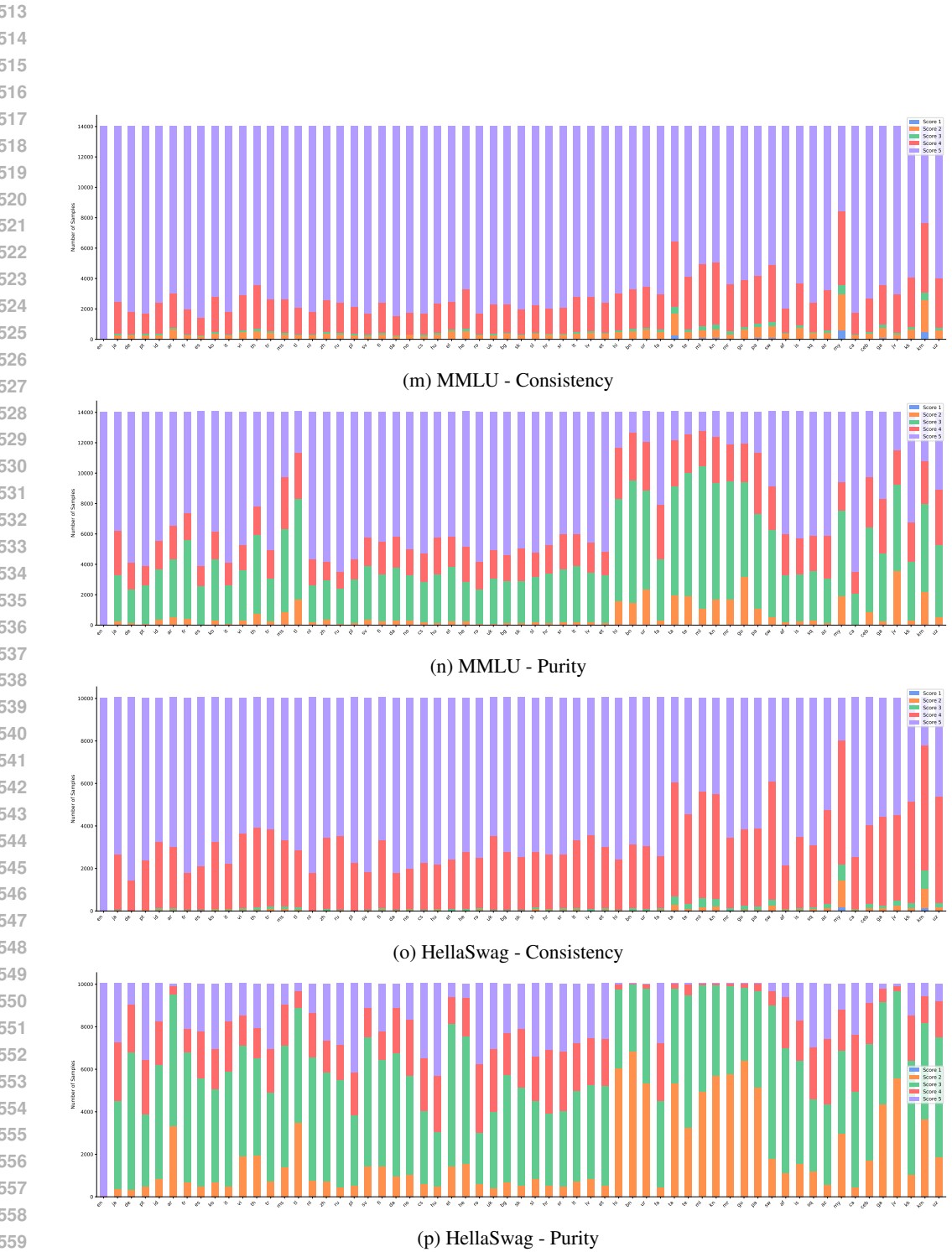

(m) MMLU - Consistency

(n) MMLU - Purity

(o) HellaSwag - Consistency

(p) HellaSwag - Purity

Figure 9: Consistency and purity distributions evaluated by GPT-4o (Part 4/6)

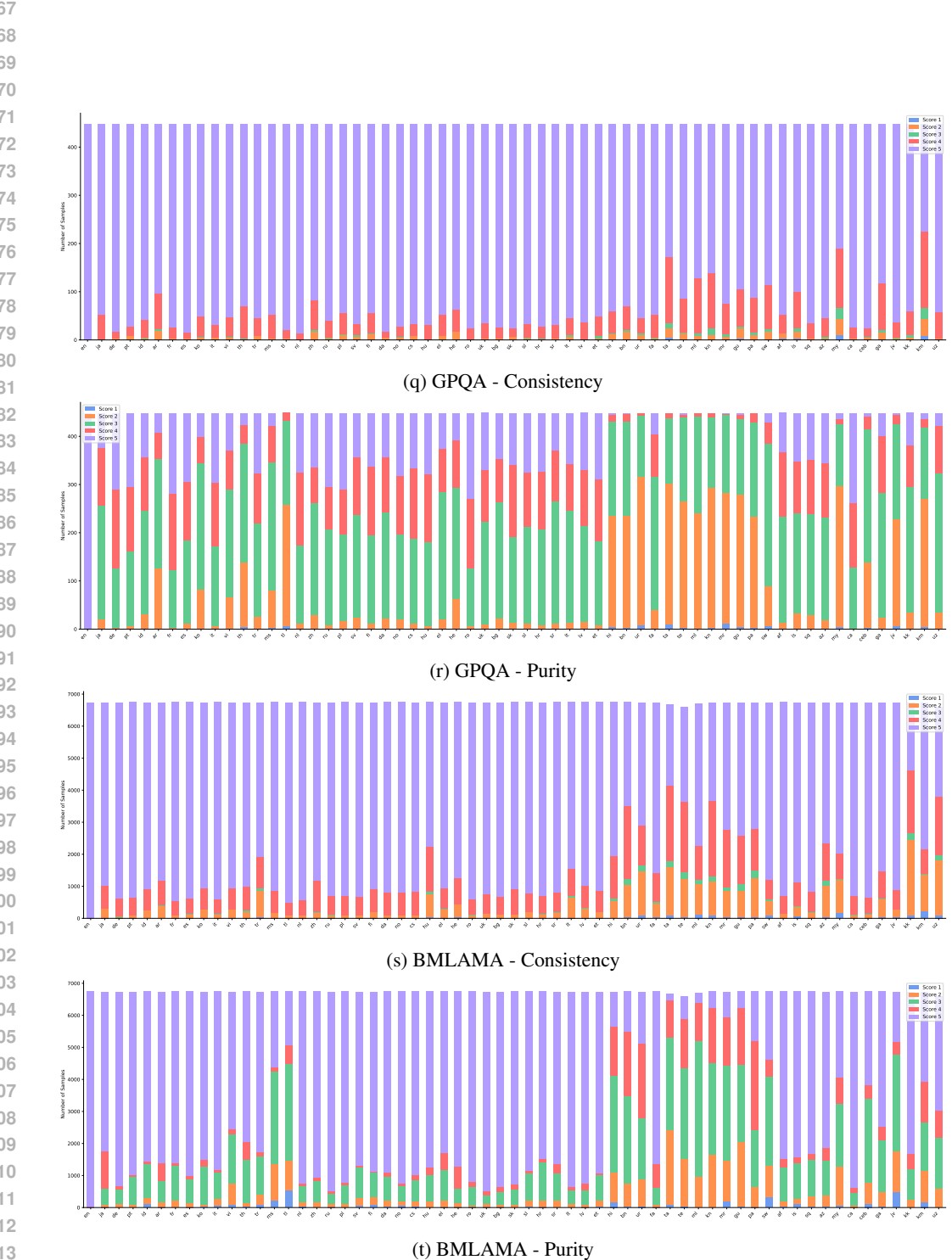

(q) GPQA - Consistency

(r) GPQA - Purity

(s) BMLAMA - Consistency

(t) BMLAMA - Purity

Figure 9: Consistency and purity distributions evaluated by GPT-4o (Part 5/6)

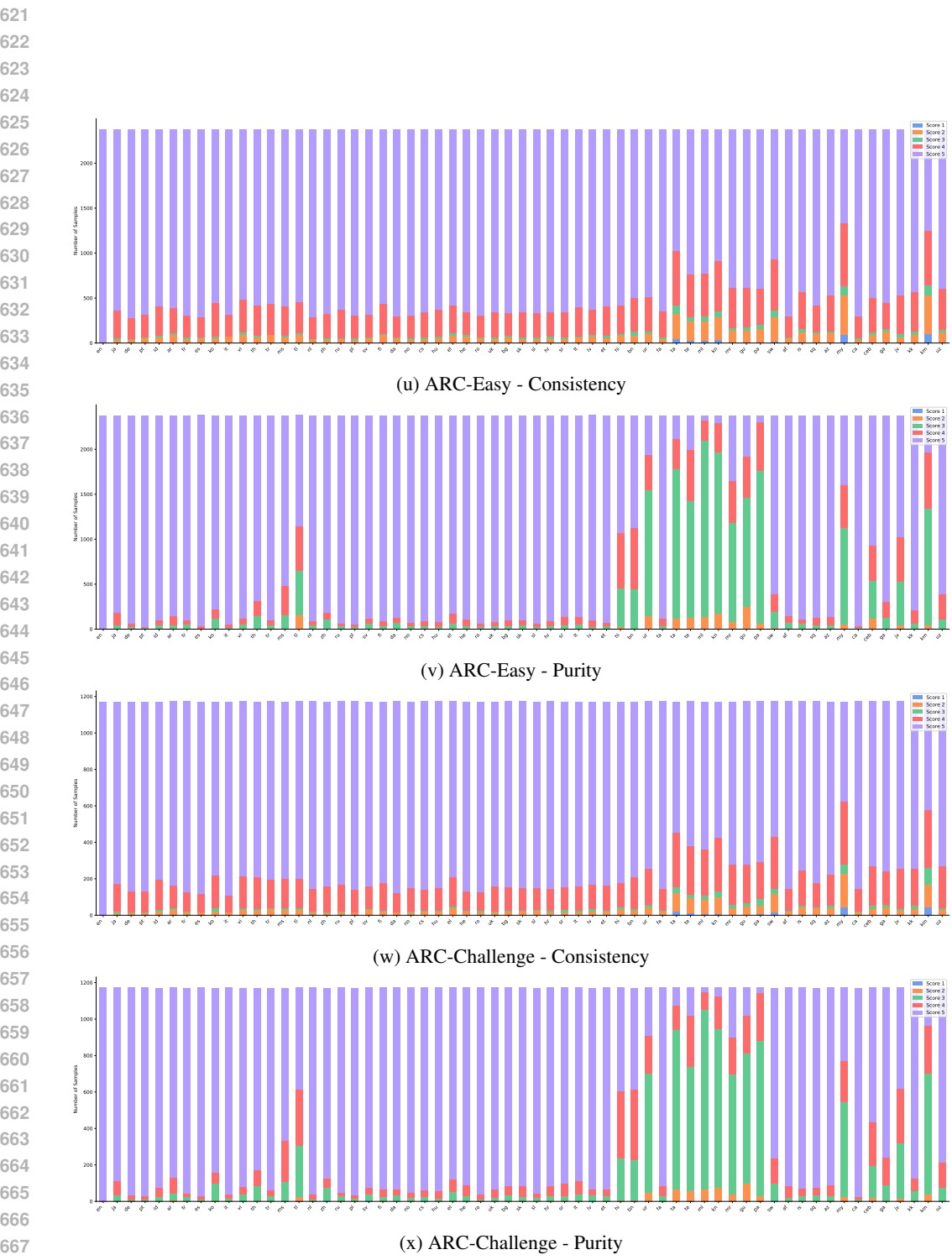

(u) ARC-Easy - Consistency

(v) ARC-Easy - Purity

(w) ARC-Challenge - Consistency

(x) ARC-Challenge - Purity

Figure 9: Consistency and purity distributions evaluated by GPT-4o (Part 6/6)

