# OpenReview forum: "MuBench: Assessment of Multilingual Capabilities of Large Language Models Across 61 Languages"
_ICLR.cc/2026/Conference — ICLR 2026 Conference Withdrawn Submission_

### Official Review · Reviewer_ZqrM · 2025-10-18

**Soundness:** 3
**Presentation:** 3
**Contribution:** 3
**Rating:** 4
**Confidence:** 5

**Summary:**

The authors create MuBench benchmark, covering 61 languages by translating existing datasets using their data collection pipeline with quality checks and perform human checks for 17 languages. They further evaluate LLMs on this benchmark. They perform cross-lingual consistency evaluation for consistent cross-lingual evaluation and analysis of knowledge transfer. They evaluate model performance under code switched contexts.

**Strengths:**

- Paper is well written and is easy to follow through.
- The authors covered a huge number of datasets and translated them to 61 languages which had high, medium, low-resource languages.
- Experimental setup is clearly explained and results are followed up by human evaluation.
- MuBench data collection pipeline looks thorough and has a lot of checks.
- Cross lingual consistency evaluation and creating code switched dataset to see performance on code switched data are great contributions

**Weaknesses:**

- I was a bit skeptical from the beginning about the translation quality but the fact that it was human-evaluated in 17 languages was reassuring. However, when I checked those 17 languages, most of them are either medium or high resource languages (61 languages in total out of which highest numbers are in low-resource languages (26)). This is a serious flaw in their paper. Ideally they should have picked an equal number of languages from high, medium, low resource for human evaluation. Existing LLMs don’t do well for low resource languages and I believe this is where the major gap is. I’d recommend authors to perform human evaluation for at least 8-10 low resource languages. This paper has got some amazing contributions and I’m willing to bump up the scores but low resource languages should be human evaluated to ensure correctness of their approach.
- COMET doesn’t support more than 50 languages and they mention explicitly on their website to evaluate at your own risk for languages not mentioned.
- The authors missed out on defining what they do in the “problematic samples check” step.
- Using GPT for classification is an overkill I believe. The authors should have used more efficient approaches for this like a classifier.
- Existing datasets like MMLU have some problems [1], how did authors get rid of those samples? I didn't see any table in the paper about this.


[1] Aryo Pradipta Gema, Joshua Ong Jun Leang, Giwon Hong, Alessio Devoto, Alberto Carlo Maria Mancino, Rohit Saxena, Xuanli He, Yu Zhao, Xiaotang Du, Mohammad Reza Ghasemi Madani, Claire Barale, Robert McHardy, Joshua Harris, Jean Kaddour, Emile Van Krieken, and Pasquale Minervini. 2025. Are We Done with MMLU?. In Proceedings of the 2025 Conference of the Nations of the Americas Chapter of the Association for Computational Linguistics: Human Language Technologies (Volume 1: Long Papers), pages 5069–5096, Albuquerque, New Mexico. Association for Computational Linguistics.

**Questions:**

- I’m sorry if I have skipped this but do authors share samples which are culturally sensitive?
- I don’t understand Section 3.3 partially. When you look at both models’ top choices, how do you ensure they are the same? Let’s say one language is Chinese, other is Spanish, how do they ensure answer “England” (for Spanish it would be “Inglaterra” and Chinese would be “英格兰”) are same?
- Typos:
  - Line 182: Translation
  - Line 280: Gemma2(?)
  - Line 307: Model(¿20B)

---

> ### Author Response · Authors · 2025-11-21
>
> Thank you for your thoughtful review and feedback on our submission. We appreciate the time and effort you invested in understanding our work. We would like to address the concerns you raised:
>
> > **Q1:** This paper has got some amazing contributions and I’m willing to bump up the scores but low resource languages should be human evaluated to ensure correctness of their approach.
> >
>
> **A1:**
>
> When selecting the languages for human validation, we prioritized **ensuring data quality for the most widely used languages**, while also taking into account **the difficulty of recruiting human annotators for each language**. Among the 17 languages we selected, we also **included several mid- and low-resource languages such as Indonesian, Thai and Tagalog**. Because MuBench is extremely large in scale, and recruiting human experts for very low-resource languages is significantly more difficult, we are still on the way to covered more languages. So far, our investment in **human evaluation** amounts to approximately **$31,212**. **We are continuing to invest further and are actively seeking collaboration with the community.**
>
> Nevertheless, we have **designed methods to monitor and control data quality in the languages that were not covered by human evaluation**. The translation quality in MuBench is ensured by combining human scoring with GPT-4o scoring under the same criteria, and then performing a reasonable extrapolation based on their comparison. In Table 3, we observe that GPT-4o tends to score more strictly than human annotators. It is important to note that the samples in Table 3 were drawn evenly across all score levels for the sake of comparison, which makes the displayed scores lower than the true overall scores of MuBench. By then examining GPT-4o’s actual scores across all languages (Table 4), we see that the quality remains consistently high even in low-resource languages. Therefore, **we conclude that MuBench does not suffer from significant quality degradation in the low-resource languages included.**
>
> > **Q2:** COMET doesn’t support more than 50 languages and they mention explicitly on their website to evaluate at your own risk for languages not mentioned.
> >
>
> **A2:**
>
> In constructing the dataset, we **did not use COMET scores as a basis for quality control**. We also observed that COMET fails on a few languages, such as CEB. This is also the motivation for back-translating into English and using GPT-4o for quality evaluation. The COMET scores reported in Table 4 are provided **only as a reference** to include a widely used translation quality evaluation method in addition to the LLM-as-judge approach.
>
> > **Q3:** The authors missed out on defining what they do in the “problematic samples check” step.
> >
>
> **A3:**
>
> Thank you for your careful review. In this step, we primarily perform **integrity checks** on the original English data, including verifying that all fields are present, labels exist, and candidate options are not empty. We will revise the wording to avoid misunderstanding.
>
> > **Q4:** Using GPT for classification is an overkill I believe. The authors should have used more efficient approaches for this like a classifier.
> >
>
> **A4:**
>
> Using an LLM-as-judge for classification offers the advantage of high **flexibility**. Building on the subject categories in MMLU, we expanded the taxonomy into over 100 fine-grained content categories, which are further grouped into 13 major categories, forming a **two-level hierarchy**. **Traditional classifiers would likely require re-annotating the data and retraining models, and their accuracy may still fall short of GPT-4o.**
>
> Since each sample only needs to be annotated once in English—without requiring separate annotations for every language—**the overall cost is negligible** compared to translation. Of course, if one aims to further reduce cost, this step **can easily be replaced by using lighter-weight open-source LLMs.**

---

> > ### Author Response · Authors · 2025-11-21
> >
> > > **Q5:** Existing datasets like MMLU have some problems, how did authors get rid of those samples? I didn't see any table in the paper about this.
> > >
> >
> > A5:
> >
> > We also noticed MMLU-Redux, which sampled 5,700 items from MMLU and identified correctness issues in approximately 6% of them. However, **the original MMLU remains the most widely used comprehensive knowledge QA benchmark**, **and MMLU-Redux reviewed less than half of the full dataset**. For this reason, we chose to use the original MMLU as the basis for our multilingual extension. **MuBench** **users may still remove the problematic samples flagged by MMLU-Redux if needed.**
> >
> > We understand that there may have been a misunderstanding regarding our “problematic samples check” step. In this step, we only examine the **data integrity** of the dataset (e.g., field completeness, presence of labels, absence of empty candidates), and we do not verify the correctness of each item’s answer.
> >
> > In addition to MMLU, we also observed that HellaSwag and BMLAMA contain a small number of correctness issues. However, the primary focus of MuBench is on establishing a comprehensive multilingual evaluation, and **all the datasets we selected are widely used benchmarks**. Therefore, at this stage, we did not re-audit the correctness of the original English benchmarks.
> >
> > > **Q6:** I’m sorry if I have skipped this but do authors share samples which are culturally sensitive?
> > >
> >
> > **A6:**
> >
> > Yes, in Figure 6 we present two examples of culturally unsuitable items. Our criteria for identifying culturally unsuitable samples are: whether the translation into the target language leads to misunderstanding due to differences in cultural background, and whether the item contains Western-centric concepts that are unlikely to be familiar to speakers of the target language.
> >
> > > **Q7:** I don’t understand Section 3.3 partially. When you look at both models’ top choices, how do you ensure they are the same? Let’s say one language is Chinese, other is Spanish, how do they ensure answer “England” (for Spanish it would be “Inglaterra” and Chinese would be “英格兰”) are same?
> > >
> >
> > **A7:**
> >
> > The candidate options are fully aligned across all languages, so a model’s choice in one language can be mapped to its corresponding version in any other language via the shared index.
> >
> > Thank you very much for your careful and thorough review. We sincerely hope you will consider the contribution that MuBench brings to the multilingual LLM community. We hope our clarifications address your concerns and answer your questions.

---

### Official Review · Reviewer_65Zp · 2025-10-21

**Soundness:** 3
**Presentation:** 2
**Contribution:** 3
**Rating:** 6
**Confidence:** 3

**Summary:**

This work introduces MuBench, a large-scale multilingual benchmark covering 61 languages and 3.9 million samples across a diverse range of tasks, including natural language understanding, factual knowledge, knowledge-based question answering, academic reasoning, and truthfulness. The evaluation framework examines three major aspects: overall performance, cross-lingual consistency (measured by the newly proposed MLC metric), and robustness to mixed-language inputs. The resulting experiments provide a more holistic understanding of multilingualism in large language models, offering valuable insights to guide future research and improvements.

**Strengths:**

- MuBench is broad in scope, spanning numerous languages, tasks, and samples.
- The dataset construction pipeline is carefully designed, incorporating checks for semantic consistency, translation purity, and cultural sensitivity. The authors further validate the reliability of the translations through expert evaluation on 34K samples and overlap verification with 100 samples from MMMLU.
- The experiments are conducted to evaluate the multilingual capabilities of various LLMs, revealing how cross-lingual consistency, and mixed-language contexts differ per-language performance.
- Overall, building such a large-scale benchmark and conducting these extensive evaluations represent a significant and commendable effort that will likely benefit the research community.

**Weaknesses:**

- The tasks in MuBench are mostly binary and multiple-choice formats, overlooking other important multilingual capabilities such as translation, summarization, and instruction following. This restricts the benchmark's overall applicability and impact.
- Some interpretive statements lack explicit numerical evidence. For instance, claims such as "Babel and Sailor2 demonstrate notable gains in their targeted language groups" or "smaller models often benefit from the presence of English in mixed-language inputs" would be stronger with accompanying statistical summaries or quantified comparisons (e.g., averaged improvements).
- Presenting the related work as a standalone section, instead of embedding it within lines 40–55 as a paragraph of the Introduction, would more clearly highlight the work's novelty.

**Minor Issues:**
- Typo: "Traslation" should be "Translation" (line 181).
- Missing period at the end of the sentence (line 242, after "in Appendix A.6").
- Citation error for Gemma2 (line 280).
- Table 1: "SC samples" should be corrected to "CS samples."
- Some indicators, and axis labels in figures are too small.

**Questions:**

- Why is Rel-MLC defined as MLC divided by mean accuracy? Since this normalization causes more accurate models to exhibit smaller Rel-MLC values, it may explain the apparent contradiction between MMLU and GPQA performance and the corresponding Rel-MLC values of GPT-4o (lines 387–399).
- How do LLMs respond to mixed-language inputs in terms of output language composition? Analyzing the languages used in outputs could shed light on why smaller models outperform their monolingual baselines, whereas larger models show the opposite trend.

---

> ### Author Response · Authors · 2025-11-21
>
> Thank you very much for the time and effort you devoted to reviewing our work, as well as for the detailed and constructive comments you provided. Below are our responses to the concerns and questions you raised.
>
> > **Q1:** The tasks in MuBench are mostly binary and multiple-choice formats, overlooking other important multilingual capabilities such as translation, summarization, and instruction following. This restricts the benchmark's overall applicability and impact.
> >
>
> **A1:**
>
> Here you may be **conflating task format with the capability** being evaluated. Multiple-choice is  a format for presenting questions; any type of ability can be assessed using a multiple-choice format. Moreover, because MuBench preserves the structured information of the original datasets, it is not limited to multiple-choice evaluation. The test items can also be used in other formats, such as generative matching (e.g., BMLAMA, SNLI, MultiNLI).
>
> Regarding coverage of capabilities, we began by selecting the six categories most essential for multilingual evaluation: Natural Language Understanding, Commonsense Reasoning, Factual Recall, Knowledge-based QA, Academic & Technical Reasoning, and Truthfulness. Each category contains several sub-tasks, resulting in a total of 12 tasks. Together, these tasks provide a foundational assessment of multilingual ability.
>
> MuBench is also designed to be easily extensible to other capability evaluations, and we will release more datasets in the future (currently we have already prepared math reasoning data such as MATH and GSM8K). Moreover, we will open-source the data framework so that interested researchers can extend as needed.
>
> > **Q2:** Some interpretive statements lack explicit numerical evidence. For instance, claims such as "Babel and Sailor2 demonstrate notable gains in their targeted language groups" or "smaller models often benefit from the presence of English in mixed-language inputs" would be stronger with accompanying statistical summaries or quantified comparisons (e.g., averaged improvements).
> >
>
> **A2:**
>
> Thank you for pointing this out; we will add the detail in the paper.
>
> *“ Compared to their base model, Qwen2.5-7B, Babel-9B demonstrates notable gains in its specifically optimized languages, such as TL (+6%), TR (+2%), SW (+12%) and UR (+5%). Similarly, Sailor2, which targets SouthEast Asian (SEA) languages, shows significant improvement in TL (+8%), CEB (+9%), JV (+10%) and KM (+8%).”*
>
> *“Smaller models often benefit from the presence of English in mixed-language inputs. Compared with monolingual evaluations, models with ≤4B parameters show a 1.2% overall improvement under mixed-language settings, whereas models with >4B parameters exhibit a 4.3% drop.”*
>
> We will also review the writing and add quantitative details in the other analysis paragraphs.
>
> > **Q3:** Presenting the related work as a standalone section, instead of embedding it within lines 40–55 as a paragraph of the Introduction, would more clearly highlight the work's novelty.
> >
>
> **A3:**
>
> Thank you very much for the suggestion. We will separate the related work into its own section in the revised version. We will also add a table to present a comparison between MuBench and existing datasets.

---

> > ### Author Response · Authors · 2025-11-21
> >
> > > **Q4:** Why is Rel-MLC defined as MLC divided by mean accuracy? Since this normalization causes more accurate models to exhibit smaller Rel-MLC values, it may explain the apparent contradiction between MMLU and GPQA performance and the corresponding Rel-MLC values of GPT-4o (lines 387–399).
> > >
> >
> > **A4:**
> >
> > Rel-MLC is the computation used for generating Figure 4. Since high ACC naturally inflates MLC, we focus on the relative values of MLC and ACC when drawing the heatmap, which makes the cross-language influence patterns more visually distinguishable. In Table 6, however, we report the absolute **MLC values.
> >
> > We understand that the dense layout in this section may have caused confusion, and we will improve both the formatting and the explanation in the revised version.
> >
> > > **Q5:** How do LLMs respond to mixed-language inputs in terms of output language composition? Analyzing the languages used in outputs could shed light on why smaller models outperform their monolingual baselines, whereas larger models show the opposite trend.
> > >
> >
> > **A5:**
> >
> > In our experiments, all open-source models are evaluated using **Cloze format.** This format concatenates the input and each candidate option, computes the PPL for each concatenation, and selects the option with the lowest PPL as the model’s answer. This evaluation format is widely used for assessing base models and has the advantage of high stability.
> >
> > During code-switched evaluation, the prompt, question stem, and candidates in the input are each randomly switched to any language with a certain probability. Therefore, the model is not actually generating any content during testing.
> >
> > Moreover, we use a fixed random seed when switching languages, ensuring that all models are evaluated on exactly the same data.
> >
> > We hope our clarifications address your concerns and answer your questions. Thank you once again for your valuable insights.

---

> > > ### Comment · Reviewer_65Zp · 2025-11-24
> > > **Official Comment by Reviewer 65Zp**
> > >
> > > Thank you for the clarification.
> > >
> > > While the responses help address several points, the paper would substantially benefit from clearer writing and stronger, more insightful experimental analysis, particularly experiments can uniquely be done with MuBench. For these reasons, I keep my score unchanged.

---

> > > > ### Author Response · Authors · 2025-11-24
> > > >
> > > > Thank you for your feedback and keeping the rating positive. We’re glad to see that we were able to address your concerns. We will actively answer any further questions you may have.

---

### Official Review · Reviewer_s7NU · 2025-10-31

**Soundness:** 3
**Presentation:** 3
**Contribution:** 2
**Rating:** 4
**Confidence:** 4

**Summary:**

The paper introduces a new multilingual LLM benchmark that is created by taking the existing popular English benchmarks and machine-translating them into 61 selected languages. The authors try to do automatic quality checks of the translation quality as well as a manual evaluation of 17 languages with native speakers of those languages. Using the final dataset, they then evaluate a large number of open-weight multilingual language models, showing that many of them perform much worse in other languages than English.

**Strengths:**

- The benchmark can evaluate language models on more than 60 languages, which can be very useful for those language communities -- as long as the translation is accurate and makes sense (more on that below)
- The authors double-check the quality of their translation pipeline with a manual inspection that involved native speakers of 17 languages.
- Each sample in the benchmark is annotated with the topic and sub-topic category, which might be very useful metadata for the future use of this dataset.
- There are many similar projects that take existing English datasets and naively translate them into many languages, without properly checking the translation quality. However, this work devises a multi-stage pipeline for automatically checking the translation quality.

**Weaknesses:**

As a native speaker of a lower-resource language, I find the machine-translated "multilingual" benchmarks somewhat troubling.

First of all, translationese is a problem even with human translation and much more with machine translation. You end up with a very specific unnatural variant of each languages that relies on English-like linguistic constructions and that might omit language features not present in English. As a result, such benchmarks give overly optimistic scores to English-centric language models that otherwise fail on properly created benchmarks made by native speakers. Thus, benchmarks like this can sometime do more harm than good.

My second point is more subjective; I would argue that cultural and local knowledge should be an inherent part of language evaluation -- does one really know Icelandic without recognizing a single Icelandic dish and not understanding any Icelandic cultural references? On the other hand, what I appreciate about this paper is that it takes this into account and tries to at least remove all cultural samples -- this is already much better than other similar papers that blindly translate mostly US-based questions. But it results in a dataset devoid of any local knowledge, which I believe only evaluates a certain aspect of multilinguality.

____

**Other weakness**:
- The translation from English is performed by GPT-4o, but the same model is then used to check the translation quality, which might leave many errors unchecked. It would be better to use different model(s) for the quality checks.
- Another related troubling thing is that the performance of GPT-4o is substantially lower on some languages compared to English (Figure 3). Since the same model was used to translat the questions from English, it indicates that the translation for those language is very poor. I would assume that the performance of GPT-4o would be consistent across languages if its translation is correct.
- You say that *"we chose the 61 most widely spoken languages based on the number of native speakers"* (lines 105--106), which is not true. For example, Hausa (with 58 million speakers) or Bhojpuri (with 53 million speakers) are not included even though Icelandic (0.3 million speakers) is included.
- Translation of some of the tasks, those that rely more on specific language features, can be problematic. For example, if you translate the WinoGrande example "My shampoo did not lather easily on my Afro hair because the _ is too dirty. (answer: shampoo / hair)" into a language like Czech (where "shampoo" and "hair" are of different grammatical genders), the sample loses any ambiguity and thus it no longer evaluates the same thing. I wonder how much is the observed performance drop across many tasks in Table 5 connected to such issues. For example, losing more than 30 percentage points on translated HellaSwag (81.5 -> 49.4) but only 7 on MNLI (88.0 -> 80.4) is slightly concerning.

**Questions:**

- One thing that I didn't understand is why the cross-lingual alignment is such an important feature for a multilingual benchmark? From my point of view, the three related works listed in the introduction -- CMMLU, ArabicMMLU and INCLUDE -- are much more useful for evaluating multilinguality as they also localize the benchmarks. But you say (line 47) that the great benefit of your benchmark is cross-lingual alignment, so why is it so important? Cannot we evaluate consistency across languages without this alignment?
- As far as I can tell, the 17 evaluated languages are fairly high-resource, wouldn't it be more interesting to more closely check the translation quality of the lower-resource tail of languages?

---

> ### Author Response · Authors · 2025-11-21
>
> Thank you very much for your valuable feedback. We can see that you reviewed the paper with great care. It is a pleasure to discuss with you.
>
> > **Q1:**
> Translationese is a problem even with human translation and much more with machine translatioon.
>
> > Cultural and local knowledge should be an inherent part of language evaluation.
> >
>
> **A1:**
>
> We appreciate your thorough consideration.
>
> First, we would like to clarify the motivation behind this work. We have observed a large number of recently released large language models claiming support for dozens or even hundreds of languages. However, their technical reports typically evaluate multilingual capability on only two or three tasks and on a very limited set of languages. Moreover, many research works on multilingual LLMs still rely on a small set of benchmarks, including several that are simply translations from English. The quality control behind these translated benchmarks is often opaque, and their language coverage is limited and inconsistent. This creates substantial barriers for multilingual research.
> **MuBench is designed to fill this gap: it aims to bring multilingual evaluation—both in scale and in language coverage—up to a level comparable to English evaluation, while ensuring clearly defined and transparent quality control.** This provides an essential foundation for multilingual research in the community.
>
> We agree with the reviewer that translation is never perfect, not even with human translators. **We have therefore explicitly designed mechanisms to control for translation quality**. During translation, every translation into a target language is back-translated into English, and we compare the semantic consistency between the original English text and the back-translated version. This ensures that no information is lost in the translation process and that quality assessment across all languages is fair and not influenced by GPT-4o’s varying multilingual capabilities. Thus, although some translations may not reach native-speaker fluency, the test items remain fully usable.
>
> **MuBench focuses primarily on evaluating general knowledge ability across different languages.** To achieve this, we explicitly remove western-centric (culturally inappropriate) samples so that the test items remain as universally applicable as possible across languages. We strongly agree that evaluating culturally localized knowledge is also an important aspect of multilingual evaluation. In fact, we are conducting a parallel line of work specifically aimed at this problem. Constructing benchmarks for localized knowledge in each language follows a methodology completely different from MuBench, which is why it is not included within this work. We appreciate that you bring substantial expertise in multilingual research, and we would be very happy to have further discussions if you are interested.
>
> In summary, MuBench provides the multilingual community with a more transparent, quality-assured, and substantially richer set of evaluation options.
>
> > **Q2:**  The translation from English is performed by GPT-4o, but the same model is then used to check the translation quality. It would be better to use different model(s) for the quality checks.
> >
>
> **A2:**
>
> During the translation process, we back-translate each translated text into English and compare the semantic consistency between the original English and the back-translated English. This ensures that **the evaluation of semantic consistency is performed entirely in English and is therefore not affected by GPT-4o’s multilingual capability.**
>
> For the translation-purity check, the task is very simple: we only need to verify whether the translated text is indeed in the target language (to ensure the model followed the translation instruction, which in practice almost never produces negative cases). **GPT-4o is fully capable of handling these straightforward verification tasks.**

---

> > ### Author Response · Authors · 2025-11-21
> >
> > > **Q3:** Another related troubling thing is that the performance of GPT-4o is substantially lower on some languages compared to English (Figure 3). Since the same model was used to translat the questions from English, it indicates that the translation for those language is very poor. I would assume that the performance of GPT-4o would be consistent across languages if its translation is correct.
> > >
> >
> > **A3:**
> >
> > **The translation quality in MuBench is ensured by combining human scoring with GPT-4o scoring** under the same criteria, and then performing **a reasonable extrapolation based on their comparison**. In Table 3, we observe that GPT-4o tends to score more strictly than human annotators. It is important to note that the samples in Table 3 were drawn evenly across all score levels for the sake of comparison, which makes the displayed scores lower than the true overall scores of MuBench.
> > By then examining GPT-4o’s actual scores across all languages (Table 4), we see that the quality remains consistently high even in low-resource languages. Therefore, we conclude that MuBench does not suffer from significant quality degradation in the low-resource languages included.
> >
> > Additionally, we would like to highlight that **a model’s overall capability in a given language does not necessarily reflect its translation ability**. Prior work has shown that translation skills can be largely independent of general language ability. Our team has made similar observations in some of our previous work as well. You may be interested in this paper [1]. We also look forward to hearing your thoughts on this.
> >
> > [1] Anonymous Authors. Translation Heads: Unveiling Attention's Role in LLM Multilingual Translation. 2025. Under peer review.
> >
> > > **Q4:** You say that *"we chose the 61 most widely spoken languages based on the number of native speakers"* (lines 105--106), which is not true. For example, Hausa (with 58 million speakers) or Bhojpuri (with 53 million speakers) are not included even though Icelandic (0.3 million speakers) is included.
> > >
> >
> > **A4:**
> >
> > Thank you very much for pointing this out. We will revisit our language list and expand coverage to the currently unsupported languages in future updates. In fact, when selecting the list of supported languages, we considered multiple factors, including the number of native speakers, the amount of that language’s tokens available in Common Crawl, and the level of support in existing multilingual datasets and tools. We will also clarify this point more explicitly in the paper.
> >
> > > **Q5:** Translation of some of the tasks, those that rely more on specific language features, can be problematic.
> > >
> >
> > **A5:**
> >
> > We were unable to locate the specific example you mentioned within MuBench. Based on the example you provided, our understanding is that the translation might actually make the item easier, which could increase the model’s accuracy rather than decrease it. When translating WinoGrande, we used the original sentences containing the underscore placeholder and asked GPT-4o to translate those directly. Therefore, the answer was never exposed to GPT-4o during translation. **WinoGrande fundamentally evaluates a model’s understanding of real-world concepts rather than grammatical cues, so the model is supposed to rely on fact rather than syntax-based heuristics to resist distractors and select the correct answer.**
> >
> > During translation, we also noticed grammar-related challenges for certain languages. For example, in Japanese, subjects or pronouns are often omitted. Because of this, we performed strict checks on the placeholder underscore to ensure that it remained properly preserved after translation. After reviewing a large number of cases, we believe that the translated WinoGrande items still retain strong usability.
> >
> > That said, our authors are certainly not experts in every language. We would greatly appreciate any feedback you may have for the languages you are familiar with.

---

> > > ### Author Response · Authors · 2025-11-21
> > >
> > > > **Q6:** One thing that I didn't understand is why the cross-lingual alignment is such an important feature for a multilingual benchmark?
> > > >
> > >
> > > **A6:**
> > >
> > > **Multilingual evaluation has several characteristics that differ from monolingual evaluation.** Single-language evaluation primarily compares model performance across models, whereas **multilingual evaluation must not only compare models but also compare a model’s relative strengths across languages**. This requires having the same test items across languages and focusing on knowledge that is universally applicable across them.
> > >
> > > In addition, **we need models to provide consistent answers to the same question asked in different languages**—i.e., cross-lingual consistency—which also requires alignment of test items across languages.
> > >
> > > Furthermore, this alignment also enables us to construct code-switched test sets, allowing us to evaluate a model’s ability to handle mixed-language contexts. All of these aspects are crucial for studying cross-lingual transfer, which is why MuBench places strong emphasis on cross-lingual alignment in evaluation.
> > >
> > > Of course, localized evaluations for each language are also an important dimension of multilingual evaluation. Because the methodology for building localized datasets differs fundamentally from MuBench, we address this in a separate concurrent work.
> > >
> > > We can see that you have extensive experience in multilingual LLM research, and we would be very glad to continue the discussion with you in the future.
> > >
> > > > **Q7:** As far as I can tell, the 17 evaluated languages are fairly high-resource, wouldn't it be more interesting to more closely check the translation quality of the lower-resource tail of languages?
> > > >
> > >
> > > **A7:**
> > >
> > > Among the 17 languages we selected, we also included several mid- and low-resource languages such as Indonesian, Thai and Tagalog. Because MuBench is extremely large in scale, and recruiting human experts for very low-resource languages is significantly more difficult, we have not yet been able to cover all languages.
> > >
> > > So far, our investment in human evaluation amounts to approximately $31,212. We are continuing to invest further and are actively seeking collaboration with the community. We would very much like to work with researchers like you, who possess rich linguistic expertise, to further improve the quality of the data.
> > >
> > > We hope our clarifications address your concerns and answer your questions. Thank you once again for your valuable feedback.

---

### Official Review · Reviewer_7SaQ · 2025-11-01

**Soundness:** 2
**Presentation:** 3
**Contribution:** 1
**Rating:** 4
**Confidence:** 4

**Summary:**

The paper introduces MuBench, a multilingual benchmark for evaluating LLMs across 61 languages covering tasks such as NLU, commonsense reasoning, factual recall, QA, and truthfulness. It emphasizes cross-lingual alignment, cultural sensitivity checks, and proposes a Multilingual Consistency (MLC) metric. The benchmark includes code-switched and multi-format (local, English-template, cloze, and mixed) variants. The authors evaluate several open and proprietary models, finding persistent performance gaps between English and low-resource languages and little improvement from model scaling.

**Strengths:**

- Ambitious scope and coverage: 61 languages and multiple task categories represent an impressive effort toward comprehensive multilingual evaluation.

- Detailed translation pipeline: The multi-stage quality control with semantic, purity, and cultural sensitivity checks is well-structured and thorough.

- Cross-lingual alignment and code-switching evaluation: Enables new analyses not possible with existing benchmarks.

- Transparency and openness: Dataset availability on Hugging Face improves reproducibility and potential reuse.

- Empirical findings: Highlights real and relevant disparities between English and non-English model performance.

**Weaknesses:**

- Limited novelty: The contribution is primarily engineering and dataset aggregation, not a clear conceptual or methodological innovation beyond existing multilingual benchmarks (e.g., MMLU, BenchMAX, INCLUDE).

- Benchmark saturation – Given numerous existing multilingual datasets, the incremental improvement offered by MuBench does not clearly justify publication in a top-tier venue like ICLR.

- Evaluation analysis lacks depth:  insight into causes or linguistic patterns, error analysis, and detailed methodological justifications are mostly missing or superficial.

- Repetition and length: The paper reads as overly descriptive and dataset-heavy, lacking theoretical framing or hypothesis-driven evaluation.

**Questions:**

as suggestion: since the paper focuses on cross-lingual alignment, the authors may also see this recent paper as well on the same topic: https://aclanthology.org/2025.findings-acl.1385/

---

> ### Author Response · Authors · 2025-11-20
>
> Thank you for your thoughtful review and feedback on our submission. We appreciate the time and effort you invested in understanding our work. We would like to address the concerns you raised:
>
> > **Q1:** Limited novelty: The contribution is primarily engineering and dataset aggregation, not a clear conceptual or methodological innovation beyond existing multilingual benchmarks.
> >
>
> **A1:**
>
> Compared with existing multilingual dataset construction methods, our data construction framework is highly automated, which reduces reliance on human annotation and enables rapid scalability. This is particularly important for multilingual evaluation, where non-English data is scarce and manual construction is costly. Given the inherent resource gap between multilingual and English data, manually assembled multilingual benchmarks have struggled to reach the scale and comprehensiveness of English evaluations. As a result, although many multilingual LLMs are being released, their technical reports often contain very limited and fragmentary multilingual evaluation.
>
> Our work fills this gap: **for the first time, we provide assessment of multilingual LLMs across a wide range of languages and capabilities,** and we further offer **new insights on mixed-language performance and cross-lingual consistency**.
>
> **In terms of the data construction framework**, our innovations include:
>
> 1. **Abstracting heterogeneous English data sources** into a unified, highly reusable pipeline;
> 2. **Introducing multiple automatic quality control metrics** during data translation to reduce reliance on human post-editing;
> 3. **Incorporating variant design** to support flexible evaluation protocols.
> 4. **Scalability** Because of the automated design, our data framework can rapidly scale to more datasets along both language coverage and data volume, which was not feasible for previous multilingual datasets that relied heavily on manual verification.
>
> > **Q2:** Benchmark saturation – Given numerous existing multilingual datasets, the incremental improvement offered by MuBench does not clearly justify publication in a top-tier venue like ICLR.
> >
>
> **A2:**
>
> We would like to highlight that current multilingual evaluation suites are not as numerous as they may appear. For example, in the Qwen3 technical report [1], although the model claims support for 119 languages, the multilingual evaluation only uses MMMLU (14 languages), INCLUDE (44 languages), and MGSM (11 languages), covering merely knowledge QA and mathematical reasoning. Therefore, the release of MuBench carries substantial importance for the multilingual community. We have done solid, foundational work, and we hope the community will recognize it so that it can drive further research and collaboration.
>
> The table below presents a comparison between MuBench and some of the latest multilingual benchmarks.
>
> | Multilingual Benchmark | Supported Languages | Ability Categories | Number of Tasks | Number of Samples | Cross-lingual Alignment | Multiple Evaluation Formats | Code-switched Evaluation |
> | --- | --- | --- | --- | --- | --- | --- | --- |
> | INCLUDE | 44 | 1 | 1 | 22,655 | ✗ | ✗ | ✗ |
> | MultiLoKo | 31 | 1 | 1 | 15,500 | ✓ | ✗ | ✗ |
> | BenchMax | 17 | 6 | 9 | 177,684 | ✓ | ✗ | ✗ |
> | **MuBench** | **61** | **6** | **12** | **3,921,751** | **✓** | **✓** | **✓** |
>
> [1] Yang, An, et al. "Qwen3 technical report." *arXiv preprint arXiv:2505.09388* (2025).

---

> > ### Author Response · Authors · 2025-11-20
> >
> > > **Q3:** Evaluation analysis lacks depth: insight into causes or linguistic patterns, error analysis, and detailed methodological justifications are mostly missing or superficial.
> > >
> >
> > **A3:**
> >
> > In this paper, we conduct extensive evaluations and analyses.
> >
> > - We first evaluate the overall performance of widely used multilingual LLMs (Table 5) and the distribution of their abilities across languages (Figure 3). Prior to our work, there had never been a comprehensive assessment of multilingual capability for SOTA models. Our results show that **many models experience significant degradation on low-resource languages, far below their claimed language coverage.**
> > - We then analyze cross-lingual consistency and uncover a stable influence graph across languages (Figure 4). **Further, through various typological analyses, we find that strong correlations typically occur within language families, whereas similarities in morphological type or word order do not lead to strong associations** (Appendix B).
> > - In addition, our code-switched evaluation reveals that larger models exhibit a greater performance gap between facing code-switch context and monolingual context. This may suggest that **smaller models retain more shared linguistic space, while larger models internally differentiate it into separate language subsystems.** This observation aligns with hypotheses from prior research [1], but here we provide a more comprehensive and systematic evaluation.
> >
> > In summary, MuBench offers new perspectives for studying multilingual capabilities. This is crucial for further research on how to improve multilingual capabilities. We hope that MuBench can support the multilingual community to foster further research and deeper analysis.
> >
> > [1] Chen, Jiawei, et al. "The rise and down of babel tower: Investigating the evolution process of multilingual code large language model." *arXiv preprint arXiv:2412.07298* (2024).
> >
> > > **Q4:** Repetition and length: The paper reads as overly descriptive and dataset-heavy, lacking theoretical framing or hypothesis-driven evaluation.
> > >
> >
> > **A4:**
> >
> > The **core challenge** in multilingual evaluation lies in the data gap between English and non-English languages. The motivation is clear. Therefore, **we present the data construction process as a central component of the paper in order to improve transparency**. We propose an automated approach to massively expand multilingual evaluation in a reproducible and extensible manner, reducing reliance on human annotation. We also **address aspects that prior work has often overlooked**—for example, assessing cultural appropriateness, supporting diverse testing protocols, and covering dimensions unique to multilingual evaluation (cross-lingual consistency and code-switched evaluation). We sincerely ask you to consider the contribution that MuBench brings to the multilingual research community.

---

> > > ### Author Response · Authors · 2025-11-21
> > >
> > > > **Q5:** as suggestion: since the paper focuses on cross-lingual alignment, the authors may also see this recent paper as well on the same topic: https://aclanthology.org/2025.findings-acl.1385/
> > > >
> > >
> > > **A5:**
> > >
> > > Thank you very much for sharing. We carefully read this paper. The authors use the degree of alignment between hidden states of multilingual sentences as a proxy for evaluating a model’s multilingual capability, and they show that this proxy correlates well with downstream task performance using Pearson scores. Although both this work and MuBench are motivated by the goal of improving multilingual evaluation, we believe the two are not parallel. Our work focuses on **building a systematic and human-interpretable evaluation framework**, whereas the MEXA work proposes **a model-internal proxy metric**. Thus, the methodological scope and contributions differ substantially.
> > >
> > > - **MEXA provides only an aggregated estimate of a model’s multilingual capability**, lacking fine-grained evaluation across languages, capability categories, and specific tasks. In contrast, MuBench is designed to support multi-perspective analyses at the language and capability coverage.
> > > - **Although the paper reports good correlations between the proxy metric and downstream task performance, a clear gap remains between the proxy and the model’s actual behavior.** Pearson scores can easily appear high on two benchmarks, yet such correlations are highly sensitive to model selection and rarely provide actionable evidence when optimizing real models. Thus, the proxy’s coarse alignment with downstream tasks lacks sufficient precision to guide model development.
> > > - **MEXA can be applied only to English-centric models.** If a model’s language-specific subsystems diverge substantially, the proxy quickly collapses. Our evaluation of contextual processing ability (Section 3.4) shows that larger models may exhibit increasingly separated internal language representations. Therefore, a pivot-language-based proxy may not be valid for all model architectures or training stages.
> > > - **The correlation validation in MEXA is limited to only three benchmarks (Belebele, m-MMLU, m-ARC)**, which **again reflects the scarcity of multilingual evaluation data**. With MuBench, MEXA could potentially offer more comprehensive and convincing analyses.
> > >
> > > We will incorporate a comparison with works like MEXA at an appropriate place in the paper. We are very interested to hear your views on this line of work. We would also be very happy to further discuss topics related to multilingual LLM with you.
> > >
> > > We hope our clarifications address your concerns and answer your questions. Thank you once again for your valuable insights.

---

### Note · Authors · 2026-01-06

I have read and agree with the venue's withdrawal policy on behalf of myself and my co-authors.